# Structure of the human NK cell NKR-P1:LLT1 receptor:ligand complex reveals clustering in the immune synapse

Jan Bláha [1,5], Tereza Skálová [2], Barbora Kalousková [1,6], Ondřej Skořepa[1], Denis Cmunt [1,7], Valéria Grobárová[3], Samuel Pazicky[1,8], Edita Poláchová [1], Celeste Abreu[1], Jan Stránský [2], Tomáš Koval[2], Jarmila Dušková[2], Yuguang Zhao[4], Karl Harlos[4], Jindřich Hašek [2], Jan Dohnálek [2] & Ondřej Vaněk [1] ✉

Signaling by the human C-type lectin-like receptor, natural killer (NK) cell inhibitory receptor NKR-P1, has a critical role in many immune-related diseases and cancer. C-type lectin-like receptors have weak affinities to their ligands; therefore, setting up a comprehensive model of NKR-P1-LLT1 interactions that considers the natural state of the receptor on the cell surface is necessary to understand its functions. Here we report the crystal structures of the NKR-P1 and NKR-P1:LLT1 complexes, which provides evidence that NKR-P1 forms homodimers in an unexpected arrangement to enable LLT1 binding in two modes, bridging two LLT1 molecules. These interaction clusters are suggestive of an inhibitory immune synapse. By observing the formation of these clusters in solution using SEC-SAXS analysis, by dSTORM super-resolution microscopy on the cell surface, and by following their role in receptor signaling with freshly isolated NK cells, we show that only the ligation of both LLT1 binding interfaces leads to effective NKR-P1 inhibitory signaling. In summary, our findings collectively support a model of NKR-P1:LLT1 clustering, which allows the interacting proteins to overcome weak ligand-receptor affinity and to trigger signal transduction upon cellular contact in the immune synapse.

Natural killer (NK) cells are innate immune lymphocytes equipped with a wide range of activating and inhibitory surface receptors, allowing them to sensitively recognize and kill malignant, infected, or other transformed cells through "missing-" and "induced-self" mechanisms and through antibody-dependent cell-mediated cytotoxicity (ADCC)[1]. In addition, NK cells also contribute to the initiation and development of the adaptive immune response, secreting several classes of cytokines, especially proinflammatory IFN-γ[1]. Interestingly, recent findings show that NK cells can even maintain a form of immunological memory[1,2], thus further highlighting the principal roles NK cells play in immunity, particularly through their receptors.

[1]Department of Biochemistry, Faculty of Science, Charles University, Hlavova 2030, 12800 Prague, Czech Republic. [2]Institute of Biotechnology, The Czech Academy of Sciences, BIOCEV Centre, Průmyslová 595, 25250 Vestec, Czech Republic. [3]Department of Cell Biology, Faculty of Science, Charles University, Viničná 7, 12800 Prague, Czech Republic. [4]Division of Structural Biology, Wellcome Centre for Human Genetics, University of Oxford, Roosevelt Drive, OX3 7BN, Oxford, UK. [5]Present address: EMBL, Hamburg Unit c/o DESY, Notkestrasse 85, 22607 Hamburg, Germany. [6]Present address: Institute of Applied Physics – Biophysics group, TU Wien, Getreidemarkt 9, 1060 Vienna, Austria. [7]Present address: Department of Oncology, Ludwig Institute for Cancer Research, University of Lausanne, Chemin des Boveresses 155, 1066 Epalinges, Switzerland. [8]Present address: School of Biological Sciences, Nanyang Technological University, Nanyang Drive 60, 637551 Singapore, Singapore. ✉e-mail: ondrej.vanek@natur.cuni.cz

NK receptors comprise two structurally divergent classes: the families of immunoglobulin-like receptors and C-type lectin-like receptors (CTLR)[3,4]. CTLRs are encoded within the natural killer gene complex (NKC, human chromosome 12), and, unlike C-type lectins, CTLRs neither bind calcium ions nor engage carbohydrate ligands[5,6]. Instead, CTLRs are known to interact with protein ligands. For example, receptors such as Ly49, CD94/NKG2, or NKG2D recognize MHC class-I like molecules[3], whereas receptors of the NKR-P1 subfamily recognize structurally highly related Clr/Ocil CTLRs. These are encoded by *CLEC2* genes[3,5] genetically tightly linked to the NKR-P1-coding *KLR* genes. This unique CTLR:CTLR interaction system is involved in both non-MHC missing-self and induced-self recognition[3–5]. Several inhibitory and activating NKR-P1 receptors have been described in mice and rats; however, the human receptor NKR-P1 (CD161, *KLRB1* gene) remains since 1994 the only human orthologue described so far[7]. Nevertheless, based on structural and functional homology to NKR-P1, the human activating CTLR:ligand pairs NKp65:KACL (*KLRF2:CLEC2A*)[8] and NKp80:AICL (*KLRF1:CLEC2B*)[9] have been proposed as the activating counterparts of human NKR-P1[4,10].

Human NKR-P1 (CD161) was first reported as a marker of NK cells[7], in which NKR-P1 acts as an inhibitory receptor[7,11,12] up-regulated by IL-12[13]. However, NKR-P1 is also expressed by natural killer T (NKT) cells[14], mucosal-associated invariant T (MAIT) cells[15], and other subsets of T-lymphocytes[16], wherein NKR-P1 acts as a co-stimulatory receptor, increasing IFN-γ secretion[11,17]. Unsurprisingly, NKR-P1 is even detected in immature CD16− CD56− NK cells[18] and in precursors of Th17 and MAIT cells in the umbilical cord blood[19]. Recently, NKR-P1 was identified in glioma-infiltrating T cells, having an inhibitory, immunosuppressive role in T cell-mediated killing of glioma cells[20]. In addition, NKR-P1 promotes transendothelial migration to immunologically privileged niches upon interaction with its endogenous ligand, lectin-like transcript 1 (LLT1)[19,21,22].

LLT1 (gene *CLEC2D*) is primarily expressed on activated monocytes and B cells[23]. In these cells, LLT1 helps to maintain NK cell self-tolerance[10,23]. However, IL-2 can induce its expression on NK and T cells[24]. Furthermore, LLT1 is up-regulated on glioblastoma[25], prostate and triple-negative breast cancer[26,27], and B cell non-Hodgkin's lymphoma[28] cells, in which LLT1 contributes to immune evasion by dampening NK cell cytotoxicity. Interestingly, increased numbers of CD161+ Th17 cells have been detected in glioma tumors[29]. Concomitantly, the functions of NKR-P1 receptors on IL-17-producing regulatory T cells[30], on subsets of Tc17 cells[31], and on all Th17 cells[19] are particularly relevant because these cells have been implicated in several autoimmune diseases (Crohn's disease[32], multiple sclerosis[33], rheumatoid arthritis[34], and psoriasis[35]). Therefore, the analysis of NKR-P1 receptors and ligands such as LLT1 is essential to gain a deeper insight into the structure-function relationships underlying both physiological and pathogenic processes in the immune system.

Human NKR-P1 and LLT1 are type II transmembrane glycoproteins with similar protein topology[4]: an N-terminal cytoplasmic signaling tail, a transmembrane helix, a flexible stalk region, and a C-terminal C-type lectin-like domain (CTLD)[3,7,36]. Moreover, both NKR-P1 and LLT1 were shown to form disulfide homodimers[7,36], likely linked in their stalk regions. However, the structure of NKp65:KACL complex[37] is the only one among all complexes of the human C-type lectin-like (CTL) receptor:ligand subfamily that has been solved so far. A subsequent study further showed that the interaction between NKp65 and KACL is protein-based and independent of glycosylation[38]. Based on these data, a model of the NKR-P1:LLT1 complex was subsequently proposed, and key interaction residues were identified through surface plasmon resonance (SPR) analysis of NKR-P1 and LLT1 mutants, highlighting the fast kinetics of this interaction[39,40]. Furthermore, the structures of related mouse NKR-P1B ectodomain complexed with murine cytomegalovirus (MCMV) immunoevasin protein m12, or with its cognate ligand Clrb, have been recently reported[41,42]. Previously,

we reported the first structure of LLT1[43] forming a non-covalent dimer regardless of glycosylation[44] and following the conserved dimerization mode of *CLEC2*-encoded ligands observed for CD69[45,46] and Clrg[47]. Notwithstanding, no comprehensive model of the CTLR:ligand complexes' dimer:dimer interaction, corresponding to the natural state of these proteins when expressed on the cells' surface, is available yet.

Here, we investigate the structure of human NKR-P1 and examine the effects of NKR-P1 dimerization on LLT1 binding. We present the crystal structure of NKR-P1 in complex with LLT1, explain the previous in-solution interaction observations, and show a novel assembly of this complex utilizing two different non-symmetric binding sites on LLT1. Our results explain how the human NKR-P1 receptor overcomes its weak affinity for LLT1 by ligand binding-induced cross-linking and clustering and elucidate the mode of signal transduction of this receptor within the NK cell immune synapse, thereby providing a good model for the future description of related homologous low-affinity complexes.

## Results

### Structure of the human NKR-P1 ectodomain

Two crystal structures of the human NKR-P1 ectodomain were solved: the structure of glycosylated NKR-P1 possessing uniform Asn-GlcNAc$_2$Man$_5$ N-glycans (NKR-P1_glyco) and of deglycosylated NKR-P1 with N-glycans cleaved off after the first GlcNAc residue (NKR-P1_deglyco); statistical data on all structures are outlined in Table 1. NKR-P1 in both crystal structures follows the general fold characteristic of a CTL domain – two α-helices (α1 and α2) and two antiparallel β-sheets with the conserved hydrophobic WIGL motif within the domain core (Figs. 1 and 2a). The two β-sheets are formed by β-strands β0, β1, β1′ and β5, and β2, β2′, β3, and β4, respectively (assignment according to Zelensky and Gready[5], also used to describe other related CTL structures[37,40]). In addition, three intramolecular disulfide bonds stabilize the domain: Cys94-Cys105, Cys122-Cys210, and Cys189-Cys202.

### The human NKR-P1 homodimer is similar to the murine dectin-1 homodimer

The asymmetric unit of NKR-P1_glyco contains two monomers, whereas the asymmetric unit of NKR-P1_deglyco contains eight NKR-P1 monomers. All these monomers are arranged into very similar homodimers, with pairwise RMSD on Cα atoms up to 0.5 Å (Fig. 2b). However, these homodimers have an unexpected configuration: they do not follow the usual dimerization mode observed for CTLDs of the *CLEC2* ligands such as CD69 or LLT1 with helix α2 at the dimerization interface. Instead, the dimerization interface of NKR-P1 is formed by helix α1, as in the murine C-type lectin-like pattern recognition receptor dectin-1 (https://doi.org/10.2210/pdb2CL8/pdb)[48], with which human NKR-P1 shares only 32% sequence identity of the CTLD (Fig. 2c). The RMSD of the Cα atoms between NKR-P1 and dectin-1 dimers is 3.7 Å in the overlapping region (matching 196 from the total 250 residues of the NKR-P1 dimer). The structurally distinct region mainly covers helices α2, whose positions differ up to 7 Å between NKR-P1 and dectin-1. We also observed a highly similar arrangement with helix α1-centered dimerization interface in the structure of a covalent disulfide dimer of rat NKR-P1B receptor ectodomain (https://doi.org/10.2210/pdb5J2S/pdb)[49] with 1.4 Å RMSD of the Cα atoms between these two dimers (Supplementary Fig. 1a). On the contrary, the non-classical dimer of mouse NKR-P1B (https://doi.org/10.2210/pdb6E7D/pdb)[42] has an entirely different overall arrangement (Supplementary Fig. 2a).

The dimerization interface of the NKR-P1 homodimer consists of six protein–protein and several water-mediated hydrogen bonds (Supplementary Table 1), a peptide bond interaction via delocalized electrons (Lys126-Glu127), and a small hydrophobic core comprising Leu119, Ala120, and Ile168 from both chains (Fig. 3). The contact surface area is ca. 500 Å$^2$. Compared to the helix α2-centered LLT1 dimer (7−12 hydrogen bonds, stronger hydrophobic core, 500−800 Å$^2$

**Table 1 | Data processing statistics and structure refinement parameters**

| Crystal structure | NKR-P1 glyco | NKR-P1 deglyco | NKR-P1:LLT1 |
|---|---|---|---|
| PDB code | 5MGR | 5MGS | 5MGT |
| **Data processing statistics** | | | |
| Space group | $P3_121$ | $P1$ | $P2_12_12_1$ |
| Unit-cell parameters a, b, c (Å); α, β, γ (°) | 68.24, 68.24, 127.19; 90, 90, 120 | 44.81, 68.40, 101.56; 101.88, 100.72, 100.64 | 44.58, 80.15, 272.95; 90, 90, 90 |
| Resolution range (Å) | 43.3–1.8 (1.84–1.80) | 48.68–1.90 (1.93–1.90) | 76.90–1.90 (1.94–1.90) |
| No. of observations | 1,287,150 (76917) | 610,421 (26224) | 1,452,154 (80494) |
| No. of unique reflections | 32,555 (1900) | 87,081 (4353) | 78,617 (4472) |
| Data completeness (%) | 100 (100) | 97.9 (95.1) | 100 (99.9) |
| Average redundancy | 39.5 (40.5) | 7.0 (6.0) | 18.5 (18.0) |
| Mosaicity (°) | 0.08 | 0.09 | 0.05 |
| Average $I/\sigma(I)$ | 41.0 (7.4) | 12.9 (1.7) | 14.0 (3.0) |
| Solvent content (%) | 47 | 42 | 57 |
| Matthews coefficient (Å³/Da) | 2.32 | 2.13 | 2.32 |
| $R_{merge}$[†] | 0.061 (0.637) | 0.085 (0.894) | 0.153 (0.976) |
| $R_{pim}$ | 0.010 (0.102) | 0.053 (0.618) | 0.052 (0.337) |
| $CC_{1/2}$ | 1.000 (0.975) | 0.999 (0.671) | 0.998 (0.890) |
| **Structure refinement parameters** | | | |
| $R_{work}$ | 0.167 | 0.157 | 0.166 |
| $R_{free}$ | 0.202 | 0.207 | 0.201 |
| $R_{all}$ | 0.168 | 0.157 | 0.166 |
| Average $B$-factor (Å²) | 33 | 32 | 26 |
| RMSD bond lengths from ideal (Å) | 0.016 | 0.018 | 0.019 |
| RMSD bond angles from ideal (°) | 1.7 | 1.8 | 1.8 |
| Number of non-hydrogen atoms | 2481 | 8344 | 7030 |
| Number of dimers per asymmetric unit (chains) | 1 NKR-P1 (AB) | 4 NKR-P1 (AB, CD, EF, GH) | 1 LLT1 (AB), 2 NKR-P1 (CD, EF) |
| Number of water molecules in the asymmetric unit | 225 | 682 | 691 |
| Positions of modeled GlcNAc residues | AB 116, AB 169 | C 116, ABCEFGH 157, ABCDEFGH 169 | AB 95, AB 147, F 116, CDF 157, CDEF 169 |
| Ramachandran statistics: residues in favored regions (%); number of outliers[82] | 98; 0 | 98; 0 | 98; 0 |

Values in parentheses refer to the highest resolution shell.

[†]$R_{merge} = \sum_h\sum_i|I_{hi} - \langle I_h\rangle|/\sum_h\sum_iI_{hi}$ $R_{p.i.m.} = \sum_h\sum_i(n_h - 1)^{(-1/2)}|I_{hi} - \langle I_h\rangle|/\sum_h\sum_iI_{hi}$, and $R = \sum_h||F_{hobs}| - |F_{hcalc}||/\sum_h|F_{hobs}|$, where $I_{hi}$ is the observed intensity, $\langle I_h\rangle$ is the mean intensity of multiple observations of symmetry-related reflections, while $F_{hobs}$ and $F_{hcalc}$ are the observed and calculated structure factor amplitudes, respectively. $R_{work}$ is the $R$ factor calculated on 95% of reflections, excluding a random subset of 5% of reflections marked as "free". The final structure refinement was performed on all observed structure factors.

contact surface area)[43], the helix α1-centered NKR-P1 dimer forms through a smaller contact surface area with fewer contact residues.

**Glycosylation of human NKR-P1 affects its dimerization**

The NKR-P1 ectodomain contains three potential N-glycosylation sites at residues Asn116, Asn157, and Asn169 (Fig. 1a). Glycosylation at Asn169 is visible in the electron density maps of both NKR-P1_glyco and _deglyco structures. In NKR-P1_glyco, the complete GlcNAc$_2$Man$_5$ carbohydrate chain is localized in chain A, whereas a partial GlcNAc$_2$Man$_3$ chain is localized in chain B (Fig. 3a). In NKR-P1_deglyco, a single GlcNAc unit remaining at Asn169 can be well-identified in all eight NKR-P1 chains. Interestingly, the localized first GlcNAc units linked at Asn116 in NKR-P1_glyco and the overlapping GlcNAc units at Asn116 and Asn157 remaining in NKR-P1_deglyco participate in dimerization contacts with residues of helices α1 and regions β2, L1, and β2' of the opposite subunit of the NKR-P1 homodimer (Fig. 3b). In NKR-P1_glyco, five hydrogen bonds between Asn116:GlcNAc and the opposite subunit stabilize the helix α1-centered homodimer (Supplementary Table 1). The contact surface area between the opposite chain of NKR-P1 and the localized GlcNAc unit is approximately 125 Å².

In NKR-P1_deglyco, density at the homodimer interface can accommodate the remaining GlcNAc unit of either Asn116 of one chain or Asn157 of the opposing chain of the NKR-P1 dimer (for details, see

Methods, Table 1 and Fig. 3c). Conversely, in NKR-P1_glyco, the first GlcNAc unit at Asn116 is well defined in electron density in both chains, A and B, of the NKR-P1 dimer, while no electron density was found for glycosylation at Asn157. This suggests that although the glycosylation on either Asn116 or Asn157 can contribute to the dimer formation, dimerization might be impaired by steric hindrance if both N-glycans are present simultaneously. To test this hypothesis, we expressed NKR-P1 S159A mutant, thus abrogating glycosylation on Asn157. The overall fold of the mutant is comparable to the wild-type NKR-P1, as assessed by CD spectroscopy (Supplementary Fig. 3a). Indeed, when analyzed by analytical ultracentrifugation, the mutant protein exhibited substantial levels of oligomeric species (Supplementary Fig. 3b), whereas the wild-type NKR-P1 ectodomain is purely monomeric[50]. Thus, glycosylation heterogeneity may affect the propensity of human NKR-P1 to dimerize and, therefore, its ability to form multimeric complexes with LLT1.

**Structure of the human NKR-P1:LLT1 complex**

The crystal structure of the NKR-P1:LLT1 complex is formed by deglycosylated NKR-P1 and LLT1 ectodomains. The asymmetric unit of the crystal contains a complex of dimeric NKR-P1 with dimeric LLT1 and an extra dimer of NKR-P1 (Fig. 4a). These NKR-P1 dimers have the same helix α1-centered dimerization interface as the structures of the

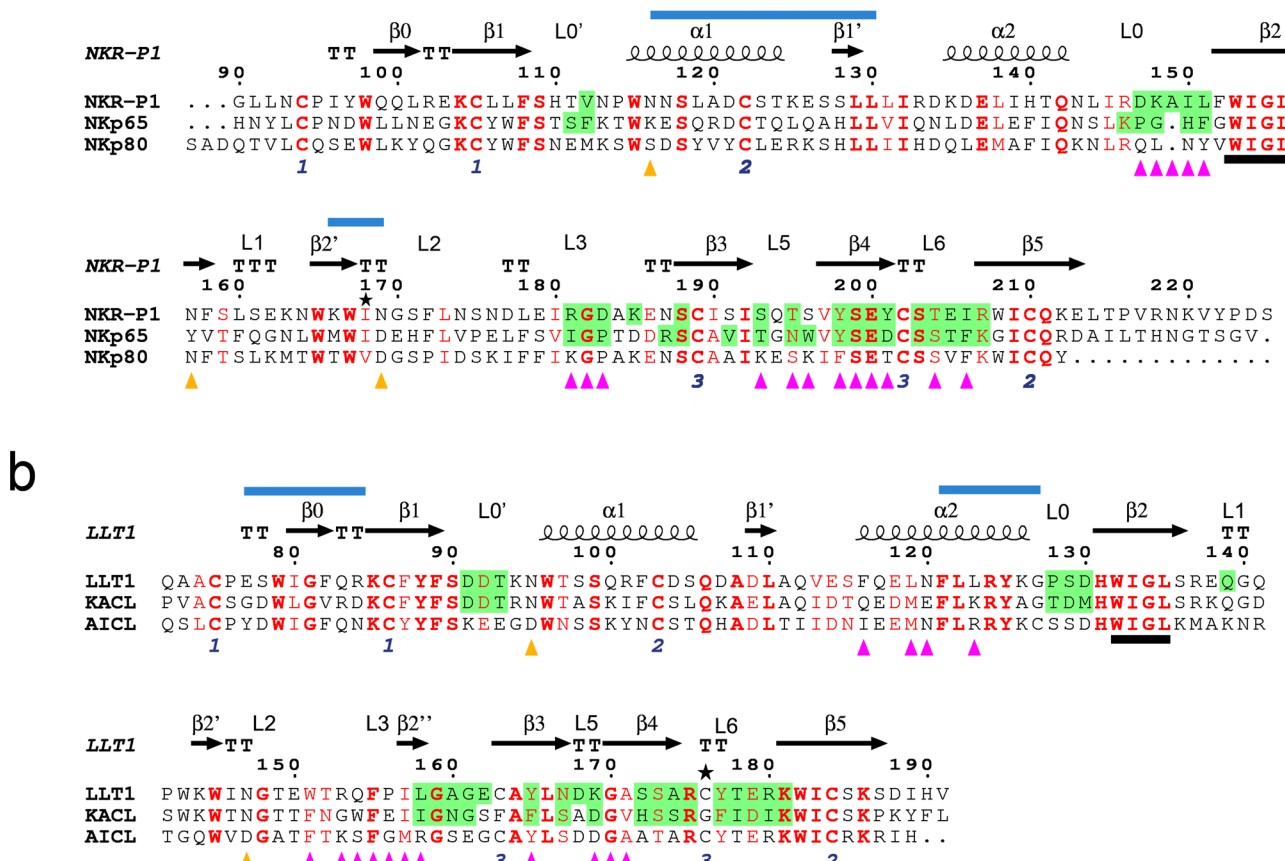

**Fig. 1 | Sequence alignment of human NKC-encoded receptor:ligand pairs showing shared structural and functional sequence motifs.** Secondary structure elements and loop regions (L) are denoted for NKR-P1 and LLT1 above the alignments. The paired numbers at the bottom indicate the disulfide pairs in the NKR-P1 and LLT1 structures; asterisks mark His176Cys mutation in LLT1 and Ile168 residue in NKR-P1. The predicted N-glycosylation sites of NKR-P1 and LLT1 are denoted with orange triangles. The conserved WIGL motifs are underlined in black. Blue lines above the sequence indicate the regions forming the non-covalent dimers of NKR-P1 or LLT1. Conserved residues are marked red; bold letters denote strictly conserved residues. **a** Sequence alignment of CTLDs of human NKR-P1-related NK cell receptors, i.e., human NKR-P1, NKp65, and NKp80. NKR-P1 residues contacting LLT1 in the NKR-P1:LLT1 complex in the primary binding mode, and NKp65 residues contacting KACL in the NKp65:KACL complex, are highlighted in green. Purple triangles indicate NKR-P1 residues that engage LLT1 in the NKR-P1:LLT1 complex in the secondary binding mode. **b** Sequence alignment of CTLDs of LLT1-related human *CLEC2* ligands, i.e., LLT1, KACL, and AICL. LLT1 residues contacting NKR-P1 in the NKR-P1:LLT1 complex in the primary binding mode, and KACL residues contacting NKp65 in the NKp65:KACL complex, are highlighted in green. Purple triangles indicate LLT1 residues that engage NKR-P1 in the NKR-P1:LLT1 complex in the secondary binding mode. The alignment was performed in Clustal Omega[83], and the graphics was prepared in ESPript 3.0[84].

unbound NKR-P1 dimers described above (Fig. 2b). The LLT1 dimer retains the expected helix α2-centered dimerization mode (Fig. 2c), identical to that previously described in unbound LLT1 structures[43]. LLT1 has clearly identifiable GlcNAc units at residues Asn95 and Asn147. The NKR-P1 glycosylation observed in the electron density of the complex matches the glycosylation identified in the NKR-P1_deglyco structure.

The LLT1 homodimer engages its partner bivalently, i.e., one dimer interacts with two NKR-P1 dimers related by crystallographic symmetry: each LLT1 monomer binds to a different subunit of a distinct NKR-P1 homodimer (Fig. 4a). There is no apparent induced fit of the binding partners – the RMSD of Cα atoms between the non-interacting and the interacting NKR-P1 dimers (NKR-P1_glyco and the complex) is 0.5 Å, and that of LLT1 (https://doi.org/10.2210/pdb4QKI/pdb and the current complex) is 0.7 Å. Moreover, the N-linked glycosylation chains do not directly contribute to the interaction.

## LLT1 engages NKR-P1 in two distinct interaction modes

NKR-P1 and LLT1 establish two types of contact in this structure – the primary (LLT1 chain B:NKR-P1 chain D) and the secondary (LLT1 chain

A:NKR-P1 symmetry-related chain C) interaction modes. The primary interaction mode matches well the structure of the homologous human NKp65:KACL complex (https://doi.org/10.2210/pdb4IOP/pdb)[37] – the RMSD of Cα atoms of the two complexes is 1.3 Å (one chain of the receptor and one chain of the ligand in each case, Fig. 4b, lower left). Similarly, in the structure of mouse NKR-P1B complexed with MCMV immunoevasin m12, the observed interaction interface matches the primary mode in the present structure of the human NKR-P1 complex, although the area covered by the m12 protein is considerably larger (Supplementary Fig. 1b, c)[41]. The recently described structure of the mouse NKR-P1B:Clrb complex (https://doi.org/10.2210/pdb6E7D/pdb)[42] also showed an interaction interface common for both NKR-P1 receptors (Supplementary Fig. 2b, c). However, the receptor:ligand arrangement observed in the second interaction mode of the NKR-P1:LLT1 complex is unique and differs from all known homologous complexes in the ligand orientation (Fig. 4b, lower right, and Supplementary Fig. 2b).

The interaction interfaces of human NKR-P1 involved in both primary and secondary interaction modes are very similar. They are formed mainly by membrane-distal residues of the L0, L3, L5, and L6

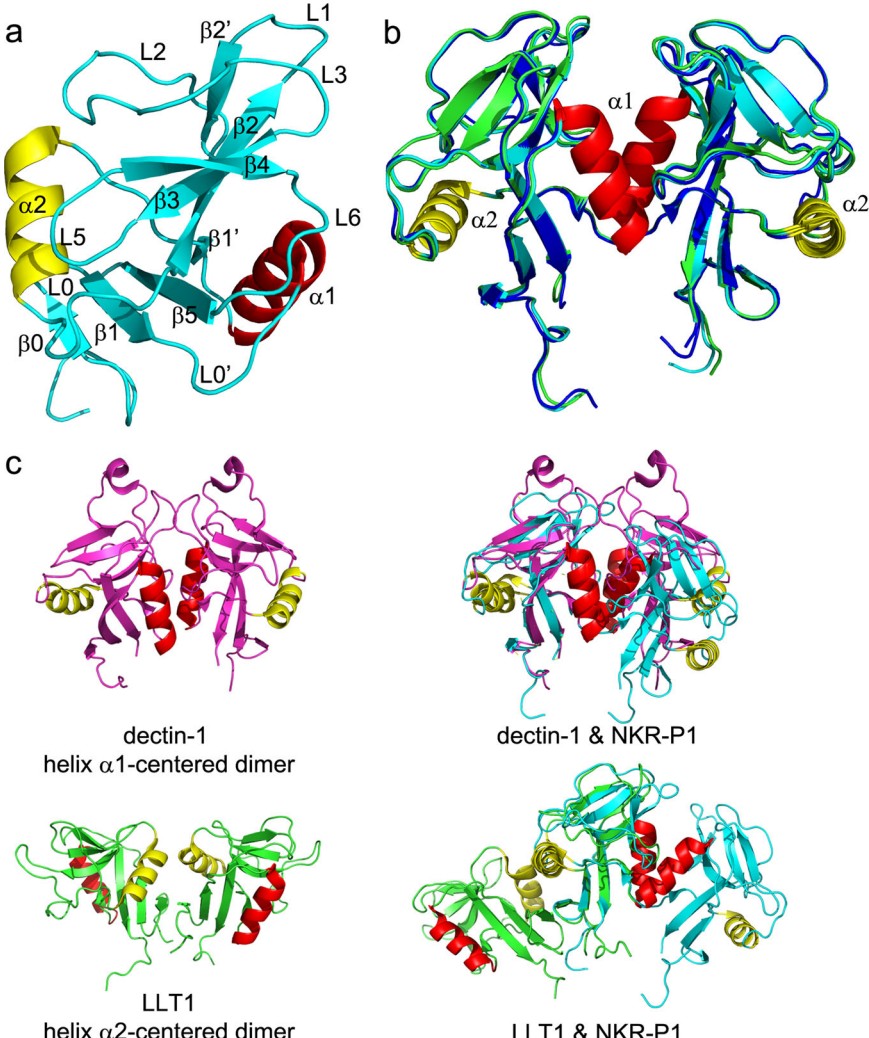

**Fig. 2 | The structure of human NKR-P1 shows a unique dimerization interface.**
**a** Ribbon diagram of the NKR-P1 CTLD. Secondary structure elements are labeled in different colors: helix α1 is red, helix α2 is yellow, and β-strands and loops are cyan. **b** Comparison between NKR-P1 dimers formed by the glycosylated (cyan), degly-cosylated free (green), and LLT1-bound (blue) forms of NKR-P1. **c** Comparison between helices α1- and α2-centered dimerization of murine dectin-1 (https://doi.org/10.2210/pdb2CL8/pdb, magenta) and human LLT1 (https://doi.org/10.2210/ pdb4QKI/pdb, green), respectively; helices α1 and α2 are shown in red and yellow. Structural alignments of dectin-1 and NKR-P1 homodimers and LLT1 and NKR-P1 homodimers, prepared by aligning only one monomer from each dimer, are shown on the right-hand side. Although the CTLD fold is conserved in each pair of the aligned monomers, the helix α1- and helix α2-centered dimers show inverse arrangement.

loops and by β3 and β4 strands (Fig. 4c, d), creating a flat surface for interaction with LLT1. By contrast, in LLT1, the primary and secondary interaction interfaces are substantially different, albeit sharing a small number of residues. While the loops L0', L0, L3, L5, L6, and strands β3 and β4 form the primary interaction patch of LLT1 (Fig. 4c), residues of the loops L2 and L5, strand β2" and helix α2 are involved in the second interaction interface (Fig. 4d). Both interaction modes place the membrane-proximal parts of the receptor and the ligand on opposite sides of the complex, creating a plausible model for interaction between two neighboring cells.

The primary interaction mode is established through nine direct and several water-mediated hydrogen bonds, in addition to two charge-supported and π-π stacking (Tyr201·Arg175) interactions, with a total contact surface area of ca. 800 Å$^2$ (Supplementary Table 1). The four strongest bonds occur between the NKR-P1 residues Arg181, Tyr201, Lys148, and Ser199 and the LLT1 residues Glu179, Glu162, Ser129, and Tyr177, respectively (Fig. 4c). The second interaction mode is established through five direct hydrogen bonds, two charge-supported and a hydrophobic (LLT1:Pro156 – NKR-P1:Ala149, Leu151)

interaction, with a total contact surface area of ca. 550 Å$^2$ (Supplementary Table 1). The three strongest bonds occur between the NKR-P1 residues Asp147, Ser199, and Arg181 and the LLT1 residues Arg153, Lys169, and Asn120, respectively (Fig. 4d).

Besides interaction with LLT1, NKR-P1 dimers bound in primary and secondary modes also have non-negligible mutual accessory contact (NKR-P1 chain D and NKR-P1 chain C in symmetry-related position, $C_{sym}$). The interface area has 440 Å$^2$ and is established through seven direct hydrogen bonds (Supplementary Table 1). Four residues are towering above the interface: Asn143 and Arg146 of chain D and Asn174 and Asn176 of chain $C_{sym}$ (none of the asparagine resi-dues is a glycosylation site). The interface also has several water-mediated contacts.

## NKR-P1:LLT1 complex formation in solution
To characterize the size and shape of the NKR-P1:LLT1 complex in solution, we have performed analytical ultracentrifugation (AUC), microscale thermophoresis (MST), and small-angle X-ray scattering coupled to size-exclusion chromatography (SEC-SAXS) experiments.

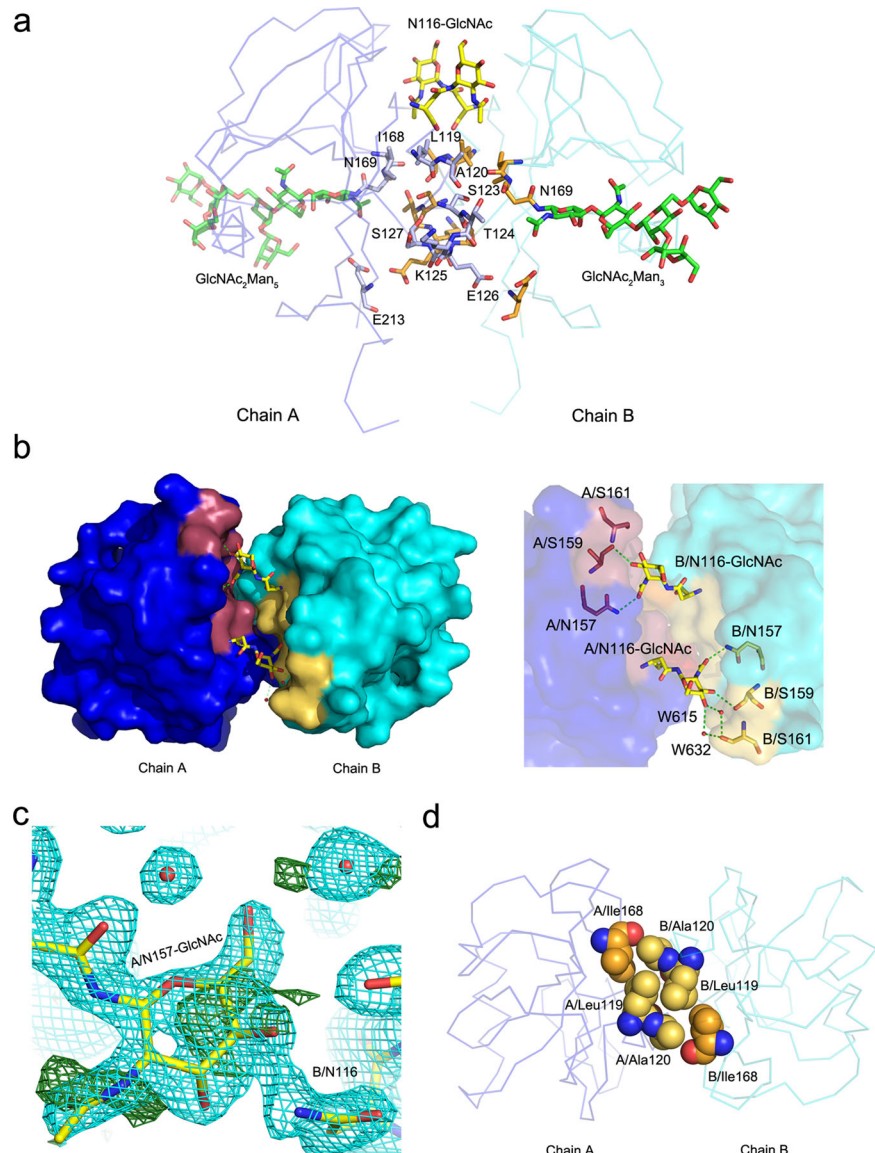

**Fig. 3 | Glycosylation affects the dimerization interface of human NKR-P1.**
**a** Dimerization interface of human NKR-P1. Subunits of human NKR-P1 are shown as Cα-trace (blue and cyan), and the dimer contact residues are shown as sticks with carbon atoms colored in light blue (blue subunit) and orange (cyan subunit); for clarity, only the residues of the blue subunit are labeled. The first GlcNAc unit N-linked to Asn116 and the carbohydrate chain N-linked to Asn169, observable in the NKR-P1_glyco structure, are shown with carbon atoms colored yellow and green, respectively. **b** Top view of the dimerization interface. The NKR-P1 subunits surfaces are colored blue and cyan. The GlcNAc units bound to Asn116 are shown as sticks with carbon atoms in yellow. Contact residues between the GlcNAc bound to chain A, and the chain B, are shown in yellow, whereas contact residues between the

GlcNAc bound to chain B, and the chain A, are shown in purple. Hydrogen bonds are shown as green-dashed lines with a detailed view on the right-hand side. **c** Mixed glycosylation states at the dimer interface in the NKR-P1_deglyco structure. The GlcNAc unit N-linked to Asn157 of chain A is modeled with an occupancy of 0.5, while the second GlcNAc unit present at Asn116 of chain B is not modeled. Contours of $2mF_o\text{-}DF_c$ (2.8σ, cyan) and $mF_o\text{-}DF_c$ (1σ, green) electron density maps are shown. **d** Small hydrophobic core in the central part of the NKR-P1 dimerization interface (subunits colored as in (**a**)). The central residues are shown as spheres with carbon atoms in yellow. The carbon atoms of Ile168 residues (whose mutation decreases the ability of NKR-P1 to bind LLT1)[51] are shown in orange.

The acquired AUC data are concentration-dependent and reflect the dynamic nature of this interacting system, similar to the results from the SEC-SAXS measurements. The free human NKR-P1 ectodomain is monomeric, with a sedimentation coefficient $s_{20,w}$ of 2.1 S corresponding to an estimated molecular mass of 18 kDa, matching the expected value of 17.5 kDa closely. The LLT1 ectodomain forms a stable non-covalent dimer (2.9 S) that does not dissociate into monomers, except in very low concentration, as previously characterized[43,50]. When increasing the loading concentration of the NKR-P1:LLT1 equimolar mixture, the sedimentation coefficient of the complex increases as well, reaching an $s_{20,w}$ value of 3.7 S at the highest analyzed

concentration (Supplementary Fig. 3c). This value, when corrected for non-ideality caused by the high protein loading concentration used (18 mg/ml), corresponds to an estimated zero-protein-concentration sedimentation coefficient $s^0_{20,w}$ value of 4.5 S, and a moderately elongated particle with approximate dimensions of 10–15 × 4–5 × 4–5 nm. That compares well with the 8–10 × 5–6 × 4–5 nm dimensions expected for the possible NKR-P1:LLT1 interaction assemblies observed in the complex's crystal structure, i.e., monomer:dimer:monomer or dimer:dimer (note that while N-glycan chains are not present in the deglycosylated complex's structure, they were present during all our analyses in solution).

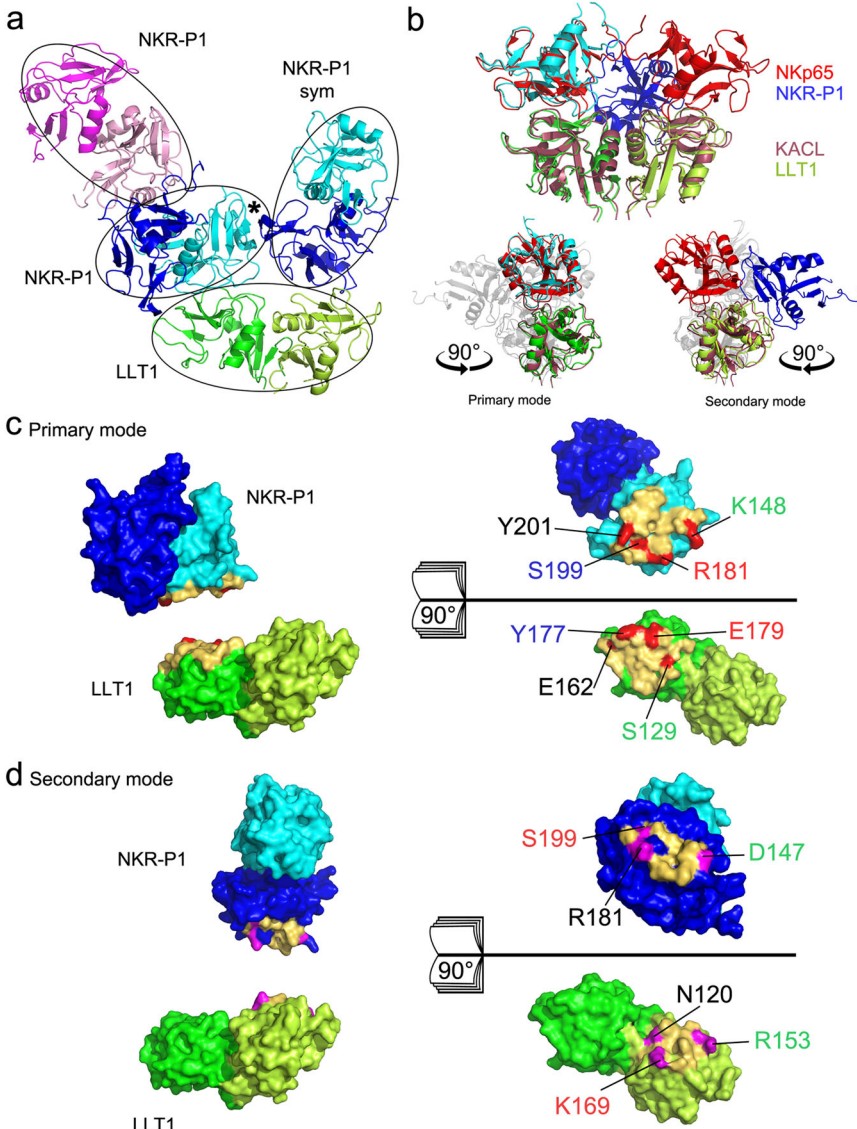

**Fig. 4 | The structure of the NKR-P1:LLT1 complex shows two distinct binding modes. a** The overall organization of the complex crystal structure. The LLT1 dimer (green/lemon) contacts the NKR-P1 dimer, formed by the blue and cyan monomers. The second blue-cyan NKR-P1 dimer is related to the first by crystal symmetry. The cyan NKR-P1 monomer interacts with LLT1 in the primary interaction mode, whereas the blue NKR-P1 monomer engages LLT1 using the secondary interaction interface. A black asterisk marks the mutual accessory contact of NKR-P1 bound in primary and secondary mode. Additionally, the asymmetric unit of the crystal contains another NKR-P1 dimer (pink/magenta) lacking contact with LLT1. **b** Overall comparison of the structure of dimeric KACL (purple) in complex with two NKp65 monomers (red; https://doi.org/10.2210/pdb4IOP/pdb) and the structure of the LLT1 dimer (green/lemon) with the two interacting NKR-P1 molecules in the primary (cyan, left side) and secondary (blue, right side) binding modes. Comparison with only the primary or secondary NKR-P1:LLT1 interaction modes is highlighted in the lower section (both in a side view, using 90° y-axis rotation). **c, d** NKR-P1:LLT1 primary and secondary interaction interfaces. Contact residues within 5 Å distance are colored in yellow. Amino acids forming the four strongest contacts are highlighted in red for the primary or magenta for the secondary mode.

## Secondary interaction mode is involved in NKR-P1:LLT1 binding in solution

To understand whether the secondary interaction mode observed within the crystal structure is also utilized in the solution, we designed an N120R, R153E, K169A triple mutant of LLT1 (LLT1$^{SIM}$), thus abolishing the three strongest contacts in the LLT1 secondary interaction interface (Fig. 4d). The LLT1$^{SIM}$ mutant was expressed and purified in the same way as the wild-type LLT1 and displayed a comparable CD spectrum (Supplementary Fig. 3a). NKR-P1:LLT1$^{SIM}$ equimolar mixture concentration series reached an s$_{20,w}$ value of 3.3 S (corresponding to an estimated s$^0_{20,w}$ value of 3.9 S), clearly showing the formation of the complex of a smaller size compared to the wild-type NKR-P1:LLT1 mixture, possibly a monomer:dimer assembly (Supplementary Fig. 3d). By integrating the whole continuous size distribution c(s)

curves and plotting the resulting weight-average S values against the LLT1 proteins' concentrations used, binding isotherms were constructed for both dilution series and firstly best-fit to the simplest hetero-association binding model A + B ⇔ AB, where A is the LLT1 dimer and B is the NKR-P1 monomer (Supplementary Fig. 4a). LLT1$^{SIM}$ showed about three-fold weaker overall affinity than the wild-type LLT1, which was further corroborated by independent MST analysis using fluorescently labeled NKR-P1 titrated with LLT1 or LLT1$^{SIM}$ (Supplementary Fig. 4b). To analyze the difference between wild-type LLT1 and LLT1$^{SIM}$ concerning the binding of the second NKR-P1 monomer, we also best-fit the data using the A + B ⇔ AB + B ⇔ BAB model (Supplementary Fig. 4c). While for the wild-type LLT1 this improved the fit and provided two different K$_D$ values, possibly corresponding to primary and secondary interaction modes, for LLT1$^{SIM}$, the fitted K$_D$ values

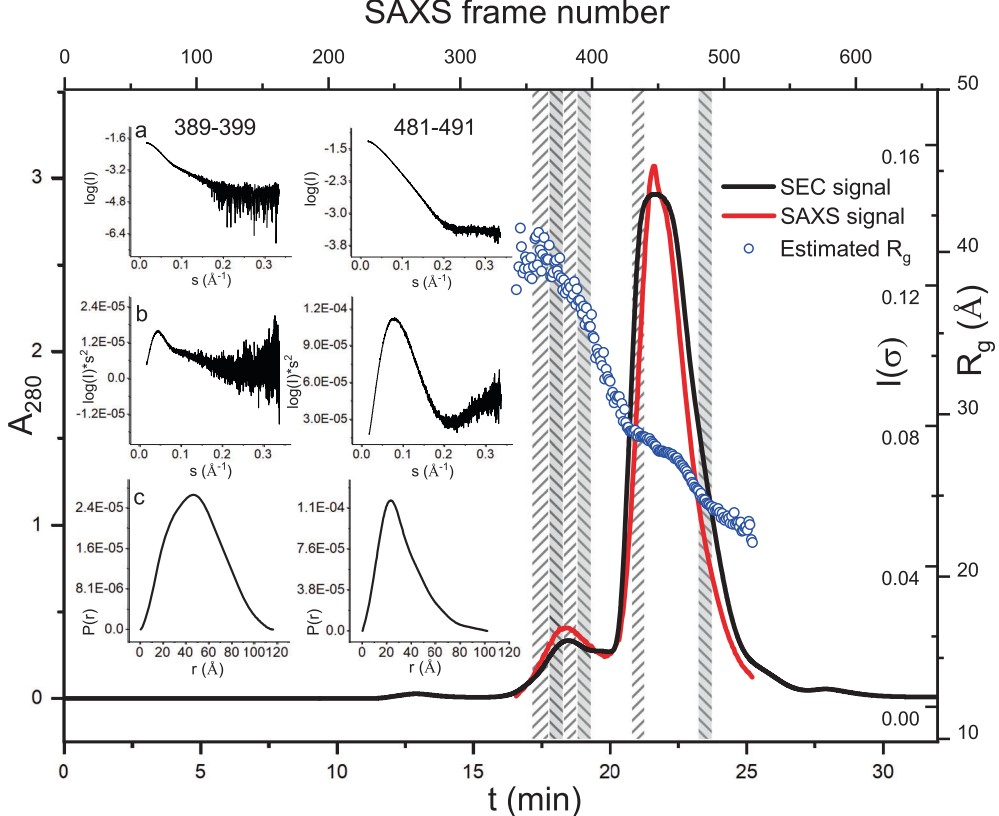

**Fig. 5 | SEC-SAXS analysis of NKR-P1:LLT1 shows higher-order complex formation.** Overlay of the size-exclusion chromatography profile (black line) and SAXS scattering signal (red line) for the NKR-P1:LLT1 equimolar mixture at a 15 mg/ml loading concentration. Both signals show two distinct peaks. For each collected SAXS frame, the radius of gyration was calculated using AUTORG (blue circles). For further analysis, six intervals of the SAXS data were selected and separately merged (frames 355–367, 368–378, 379–388, 389–399, 431–440, 481–491; denoted as columns with diagonal hatching). SAXS scattering curve (**a**), Kratky plot (**b**), and pair distance distribution function (**c**) for the intervals 389–399 and 481–491 are shown in the inset for data quality assessment of the merged data. All six merged data intervals were then individually analyzed by OLIGOMER (Supplementary Data 1 and Supplementary Fig. 5). Diffraction data from the SEC-SAXS experiment have been deposited (https://doi.org/10.17632/268ww2m4j3.1)[81].

remained unchanged, disproving the binding of the second NKR-P1 monomer. Similarly, when the AUC and MST data were globally fitted together with the AUC parameters fixed at the previously best-fit values, the LLT1$^{SIM}$ MST data could be fitted equally well with both AB and BAB models. In contrast, the fit is poor for the AB model in the case of wild-type LLT1 (Supplementary Fig. 4d). Taken together, our data show that the secondary interaction interface is not just a crystal contact but contributes significantly to the overall affinity of the NKR-P1:LLT1 interaction.

## SEC-SAXS analysis confirms NKR-P1:LLT1 higher-order complex formation

Observing assemblies of even higher order is unlikely in the AUC sedimentation velocity experiment, given the fast kinetics of this interaction, the long time-scale of the experiment, and the fact that protein concentration steadily decreases along the sedimentation boundary. However, the NKR-P1:LLT1 complex's crystal structure suggests a chain-like arrangement of receptor:ligand dimers. To determine if such assemblies exist in the solution, we performed a SEC-SAXS analysis of the NKR-P1:LLT1 equimolar mixture at a 15 mg/ml loading concentration. Two peaks of absorbance at 280 nm, corresponding to two distinct SAXS peaks, were observed in the SEC-SAXS experiment (Fig. 5). For further analysis, we sampled four data intervals from the first peak and two data intervals from the second peak and separately merged the data from within these intervals (Fig. 5 and Supplementary Fig. 5). The radius of gyration calculated from the SAXS signal steadily decreases with the retention volume. Two small

plateaus in the main peak suggest a dynamic balance between complex formation and dissociation (Fig. 5). As a result, the merged scattering curves could not be fitted well to scattering curves simulated from any single structure of the NKR-P1:LLT1 complex or its components. Therefore, we constructed models of assemblies of the NKR-P1:LLT1 complex in varying lengths (number of protein chains), considering all possible permutations of the order of the molecules using only primary, only secondary, or both interaction modes in an alternating fashion. The resultant library of all these models (in total, 49 different structures) was then used by the OLIGOMER software to best-fit the sampled data intervals (Supplementary Fig. 5) as well as the individual SAXS experimental curves (Supplementary Data 1). By this approach, the sampled SAXS data fit reasonably well to the superimposed calculated scattering curves of groups of NKR-P1:LLT1 complex structures of higher stoichiometry ($x^2$ 1.29–3.47; note that complete glycosylation was not modeled, but it contributed to the scattering; Supplementary Fig. 5). Following the continuously decreasing radius of gyration calculated from the SAXS data, the stoichiometry of the best-fit structures of the NKR-P1:LLT1 complex also constantly decreases with the retention volume. The SAXS scattering curve of the first peak is best-fit with two to three LLT1 dimers interacting in a chain-like oligomer with two to three NKR-P1 dimers, the curve of the second peak to one LLT1 dimer interacting with one NKR-P1 dimer in primary or secondary mode. Notably, OLIGOMER generally favored assemblies that combine primary and secondary interaction modes against models with the primary or secondary modes alone.

## LLT1 ligation induces NKR-P1 receptor clustering

To answer the biological relevance of these higher-order complexes in the context of a live cell, we performed single-molecule localization microscopy studies with an NKR-P1-expressing cell line and quantified the surface distribution of NKR-P1 with respect to LLT1 or LLT1$^{SIM}$ binding. Full-length NKR-P1 transfectants generated using the piggyBac system were induced to express the receptor in a limited density to allow single-molecule localization microscopy. Cells were then incubated in the presence or absence of soluble LLT1 or LL1$^{SIM}$, fixed, and labeled with anti-NKR-P1 AlexaFluor® 647 mAb. Direct stochastic optical reconstruction microscopy (dSTORM) images were acquired, and Voronoi tessellation cluster analysis was used to assess the effect of soluble LLT1 on the nanoscale organization of NKR-P1 on the cell surface (Fig. 6a). In the absence of LLT1, we detected clusters of fluorescent events with an average area of $1870 \pm 777$ nm$^2$ (Fig. 6b) and an average diameter of $41 \pm 7$ nm (Fig. 6c), mostly clusters of events with a diameter ranging from 10 to 40 nm (Fig. 6d), indicative of AlexaFluor® 647 doubly labeled NKR-P1 homodimers (ca. 6 nm + 10 nm linking error per each mAb ±15 nm of localization precision; please note that the apparent size of fluorescent clusters does not directly correspond to the size of receptor clusters). The addition of soluble LLT1 significantly increased the observed cluster of events diameters (to $47 \pm 6$ nm; $p < 0.0001$, Fig. 6c, d), the area ($2713 \pm 1180$ nm$^2$; $p < 0.0001$, Fig. 6b), and the number of events detected in the clusters ($38.8 \pm 12.3$; $p < 0.0001$, Fig. 6e), whereas the density of events in clusters remained unchanged, as expected (Fig. 6f). Furthermore, the effect of LLT1$^{SIM}$ addition on NKR-P1 nanoscale organization was statistically indistinguishable from the negative control while being significantly different from the effect of LLT1 addition. Based on the relative comparison of these three distinct receptor states (free, LLT1-bound, and LLT1$^{SIM}$-bound), we can assume that the secondary interaction interface in LLT1 is necessary to form nanoscale NKR-P1 clusters upon interaction. No significant difference in the total density of events per inner cell surface (evaluated area) was found (Fig. 6g), pointing to stable expression levels of NKR-P1 throughout the measurement. Consequently, the size of NKR-P1 nanoclusters increases (beyond one homodimer unit) not because of differences in NKR-P1 expression levels but because LLT1 cross-links two or more NKR-P1 homodimers.

## LLT1 secondary interaction mode is required for NKR-P1 inhibitory signaling

Next, we analyzed the effect of soluble LLT1 or LLT1$^{SIM}$ on the inhibitory potential of native NKR-P1 expressed on the surface of freshly isolated NK cells in an NK cell-mediated cytotoxicity assay thus investigating further the influence of NKR-P1 ligand binding-induced cross-linking on cellular signalization. NK cells isolated from the blood of three different healthy donors were activated with IL-2 and mixed with the K562 target cells in a 40:1 effector:target cell ratio and incubated for 4 h with PBS buffer as a negative control or with the soluble LLT1 or LLT1$^{SIM}$ (both in two different concentrations). In the absence of NKR-P1 ligand, K562 cells are well-susceptible to NK cell-mediated lysis (Fig. 7; PBS control, less than 10% live K562 cells). As expected, the lysis was substantially blocked in the presence of soluble LLT1. By contrast, the LLT1$^{SIM}$ variant with a mutated secondary interaction interface could not block NK cell-mediated cytotoxicity. As observed by dSTORM on the cell surface (Fig. 6), LLT1$^{SIM}$ does not efficiently cross-link NKR-P1. This mutant also fails to signalize via the NKR-P1 inhibitory receptor pathway. Therefore, based on both microscopy and cytotoxicity assay data, we conclude that NKR-P1 cross-linking triggered by LLT1 ligation is biologically relevant and indispensable for cellular signalization and requires interaction in both primary and secondary interaction modes.

## Discussion

Currently, there are only two similar CTL:CTL NK cell receptor:ligand complexes with known 3D structure: human NKp65:KACL[37] and mouse NKR-P1B:Clrb[42]. These homologous complexes are topologically similar to the primary interaction mode of the NKR-P1:LLT1 complex (Fig. 4b). Sequence identity of the extracellular part of NKR-P1 with NKp65 is 33% and with murine NKR-P1B 42%. In comparison, sequence identity of the extracellular portion of LLT1 with KACL is 49% and with murine Clrb 51%. The structure of the human NKp65:KACL complex comprises two monomeric NKp65 units interacting separately and symmetrically with a dimeric KACL ligand. Crystal structure of the murine NKR-P1B:Clrb complex shows two Clrb dimers interacting with one NKR-P1B dimer placed between them (each Clrb dimer interacts with one of the two NKR-P1B chains). The arrangement of the murine complex is similar to the herein presented NKR-P1:LLT1 complex's structure, apart from the fact that the murine complex is symmetric while we observe two distinct binding interfaces, the primary and the secondary interaction mode of LLT1.

Despite different oligomeric forms of NKR-P1:LLT1, NKR-P1B:Clrb, and NKp65:KACL complexes, the primary receptor:ligand interaction is similar in all three cases at the level of monomer:monomer superposition – the receptor:ligand pair of monomers overlaps basically along the whole chain. The most structurally conserved regions are generally sequentially conserved β-sheets in the core of the proteins. Supplementary Fig. 6a shows in red the fragments with conserved 3D position, amino acid type, and length of at least three amino acids. Such criteria are fulfilled by these fragments: in ligands – KCFYFS (in human LLT1 residues 85–90), NWT (95–97), WIGL (132–135), WKW (143–145), and WICSK (182–186), and in receptors – WIGL (in human NKR-P1 residues 153–156) and ICQ (209–211). Therefore, it seems that for their interaction, the mutual orientation of the receptor and ligand scaffold is more important than the actual interaction interface.

We noted only three amino acid interaction pairs conserved in the interaction interface in at least two of the three homologous complexes. They are gathered around residues Tyr165-Tyr171-Phe148, Arg175-Arg181-Arg158, and Tyr177-Tyr183-Phe160 of the ligand (Supplementary Table 2 and Supplementary Fig. 6a). The arginine residues form equivalent hydrogen bonds in Clrb and KACL cases. In LLT1, the arginine assumes a different conformation and forms an intramolecular hydrogen bond with Asn183. The primary interaction mode in NKR-P1:LLT1 mainly relies on main chain contacts that enable fast $k_{on}/k_{off}$ kinetics, thus corroborating the previously published SPR-based findings[39,40] and our AUC analysis. Hence, a topologically similar complex is formed in all three cases, although the underlying intermolecular recognition mechanism is semi-independent of the actual fold and amino acid composition (Supplementary Fig. 6b). However, the secondary interaction mode is unique to the NKR-P1:LLT1 complex and is not found in the other related complexes.

Y. Li and coworkers reported that the orientation of NKp65 bound to its ligand precludes the putative α2-centered dimerization of NKp65[37]. Similarly, a hypothetical NKp65 α1-centered dimer is also implausible based on steric hindrance and the lack of stabilizing interactions. This observation contrasts with the α1-centered dimerization of NKR-P1 present in both its unbound and complexed crystal structure. Interestingly, the single-nucleotide polymorphism (SNP) c.503 T > C of the human *KLRB1* gene, causing the substitution of isoleucine 168 for threonine in the NKR-P1 CTLD, was reported to have a 37% frequency of the Thr168 allele[51]. The authors showed that the Thr168 isoform of NKR-P1 has a lower affinity to LLT1 and a weaker inhibitory effect on NK cells. They proposed that Ile168 forms a part of the interaction interface between NKR-P1 and LLT1, directly affecting LLT1 recognition by NKR-P1[51]. However, the structure of the NKR-P1 homodimer shows that Ile168 is found at the dimerization interface rather than at the membrane-distal interaction interface – more specifically, in a small hydrophobic pocket within the dimerization

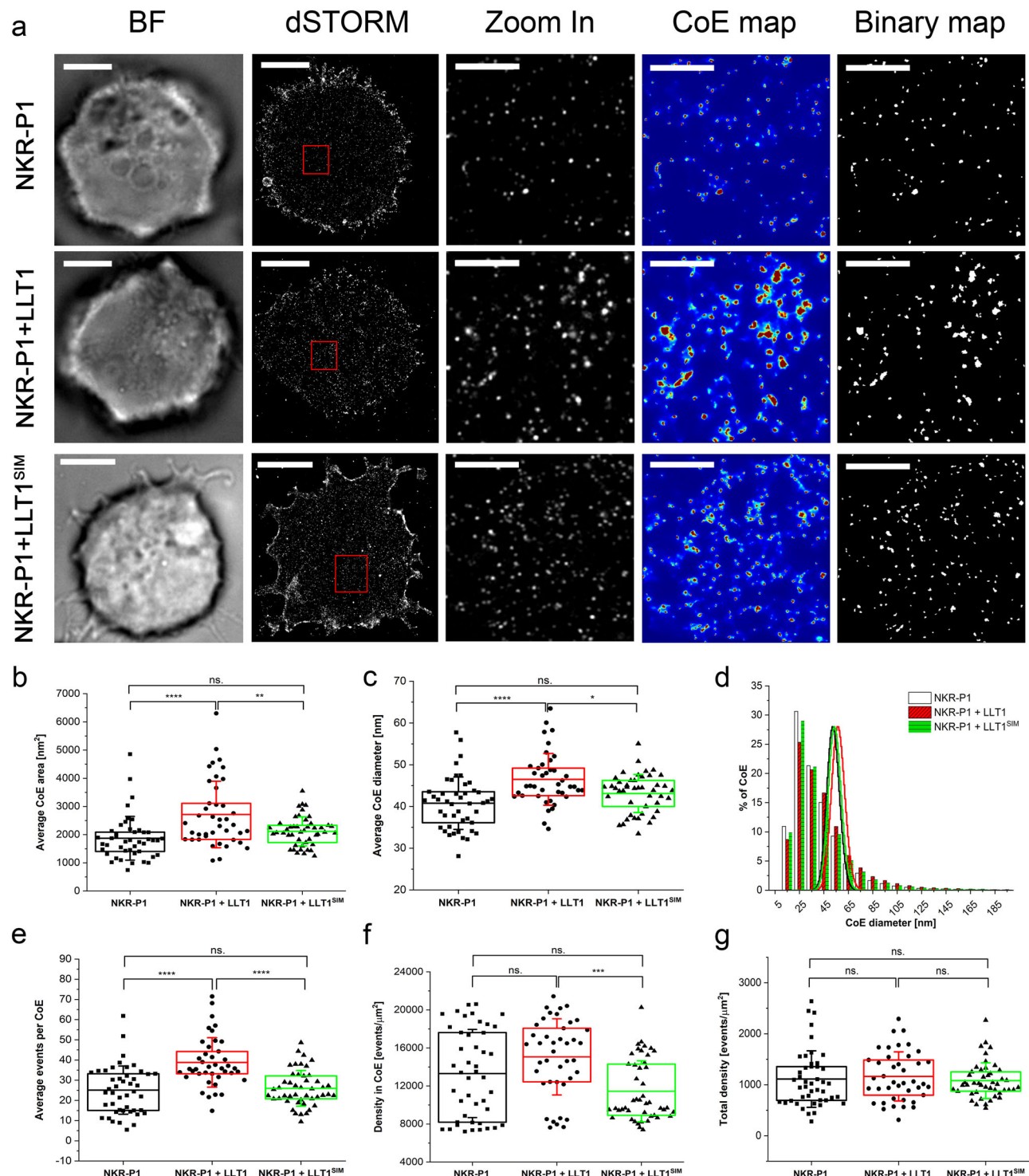

**Fig. 6 | Soluble LLT1 affects NKR-P1 distribution on the cell surface.** NKR-P1 stable transfectants were incubated in the presence or absence of soluble LLT1 or LLT1$^{SIM}$, and the cell surface distribution of NKR-P1 was monitored by super-resolution microscopy. **a** Representative brightfield (BF) and dSTORM images of full-length NKR-P1 HEK293 stable transfectants on PLL-coated slides incubated without (black) or with LLT1 (red) or LLT1$^{SIM}$ (green), fixed and stained with Alex-aFluor® 647-labeled anti-NKR-P1 mAb; scale bars represent 5 μm. The 10 μm$^2$ regions (red boxes in dSTORM images) are magnified and shown with corresponding clusters of fluorescent events (CoE) maps and binary maps; scale bars represent 1 μm. **b–g** Analysis of full-length NKR-P1 distribution relative changes induced by the presence of its LLT1 or LLT1$^{SIM}$ soluble ligands: average cluster of events area (**b**),

average cluster of events diameters (**c**), size distributions of cluster of events diameters overlaid with Poisson distribution functions (**d**), average events per cluster of events (**e**), density of events detected per cluster of events (**f**) and total density of the detected events (**g**). In **b**, **c** and **e**, **f**, each plotted point represents the mean value obtained from the analysis of the total inner surface of a single cell. The box plot center represents the overall mean value, the bounds of the box represent the interquartile range, and the whiskers represent ±SD. Data are from $n = 45$ NKR-P1$^+$ control cells and $n = 41$ or $n = 47$ NKR-P1$^+$ cells incubated with LLT1 or LLT1$^{SIM}$ in seven or four independent experiments, respectively. One-way ANOVA with Bonferroni correction, $*p < 0.05$, $**p < 0.01$, $***p < 0.001$, $****p < 0.0001$, ns. not significant. Source data are provided as a Source Data file.

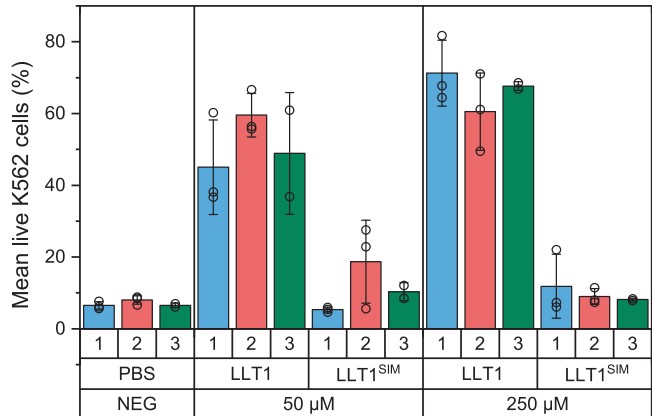

**Fig. 7 | Inhibitory effect of soluble LLT1 on the NK cell-mediated cytotoxicity.** NK cells from three different donors (blue, red, and green) and K562 target cells were incubated with PBS buffer only (negative control) or with the soluble LLT1 or LLT1[SIM] proteins, both in 50 and 250 μM concentrations corresponding to the 1× and 5× $K_D$ values, respectively, as analyzed by the AUC for the primary interaction mode (cf. Supplementary Fig. 4). Bar charts represent the means of live K562 cells in each condition with results of individual experiments plotted as empty circles, and the whiskers represent ±SD. When applicable, data were statistically evaluated by one-way ANOVA. Considering $p < 0.05$ as statistically significant, the inhibitory effect of soluble LLT1 significantly differs from LLT1[SIM], which does not differ from the control condition lacking any NKR-P1 ligand. Source data are provided as a Source Data file.

interface (Fig. 3d). Therefore, we propose that substituting the non-polar isoleucine residue with polar threonine caused by c.503 T > C SNP indirectly affects the binding affinity because this substitution destabilizes the α1-centered NKR-P1 homodimer. Glycosylation often significantly impacts receptor homooligomerization, as recently evidenced for, e.g., NK cell activation receptor NKp30[52]. NKR-P1 homodimerization is also regulated by its glycans; specifically, the glycans present on Asn116 and Asn157 (Fig. 3b). Core glycan chains present at these residues contribute partially to the α1-centered dimerization interface, but at the same time, the glycans clash together. As a result, the stability of the α1-centered NKR-P1 homodimer is improved by abrogating N-glycosylation on Asn157, leaving only the Asn116 glycan at the dimer interface (Supplementary Fig. 3b). Interestingly, a c.470 A > G SNP causing N157S mutation is also listed in the human genome variation database. However, the clinical significance of N157S mutation was not yet investigated, although it could significantly affect NKR-P1 signalization via stabilizing its ligand-bound state. The oligomeric state of the receptor might thus modulate the overall NKR-P1:LLT1 binding affinity. The NKp65:KACL complex stands out for its high affinity ($K_d \sim 0.67$ nM)[37] – ca. 3000× stronger than that of NKp80:AICL ($K_d \sim 2.3$ μM)[9] and 70,000–130,000× than that of NKR-P1:LLT1 ($K_d \sim 48$ μM[39]; this study 90 μM). Due to the exceptionally high NKp65:KACL binding affinity, any putative ancestral α1-centered dimerization interface may have been lost in NKp65. In contrast, the NKR-P1 and NKp80 receptors may have evolved to compensate for their low affinity to their ligands by utilizing the α1-centered dimerization and enabling an increased avidity effect.

The SPR analysis of single-residue mutants of both binding partners, performed by J. Kamishikiryo and updated by S. Kita and their coworkers based on the published LLT1 structure[39,40], identified several key residues of NKR-P1 and LLT1 essential for their interaction and proposed several pairs of interacting residues (Supplementary Table 3). The mutated residues that had detrimental or moderate adverse effects on binding in these SPR studies are mainly found in the primary interaction interface (Fig. 1 and Supplementary Table 3) – in LLT1: Tyr165, Asp167, Lys169, Arg175, Arg180, and Lys181, and in

NKR-P1: Arg181, Asp183, Glu186, Tyr198, Tyr201, and Glu205. However, the proposed interaction pairs do not always agree with the mutual orientation of both proteins observed in the present crystal structure of the NKR-P1:LLT1 complex. For example, the suggested LLT1/NKR-P1 pairs Tyr177/Tyr198 and Arg175/Glu200 correspond well with our observed interaction pairs LLT1:Tyr177:OH/NKR-P1:Ser199:O and LLT1:Arg175:N/NKR-P1:Glu200:OE2, respectively. On the other hand, the proposed pairing of Glu179 from LLT1 loop L6 with Ser193 and Thr195 from NKR-P1 loop L5 only resembles the crystal structure contacts of Glu179 with Arg181 (loop L3) and Tyr198 (strand β4), located near NKR-P1 loop L5. In contrast to the SPR studies, we observed no contact between Tyr165 and Phe152, although LLT1:Tyr165 is used in the primary interface, and Phe152 is close to the NKR-P1 L0 interaction region. Lastly, we cannot confirm the suggested direct bond between LLT1:Lys169 and NKR-P1:Glu205. Although these residues are close to each other in the primary mode (the closest distance of 4.3 Å), they clearly do not form a pivotal bond in the interaction interface. Furthermore, the Lys169 side chain is rather flexible, as suggested by the low quality of its electron density map in the primary mode, in contrast to all other nearby side chains with well-defined positions.

Because the secondary binding mode observed in the NKR-P1:LLT1 structure involves a different region of LLT1 than the region used in the primary binding mode, the orientation of LLT1 and NKR-P1 in this binding mode does not match the interaction pairs proposed in the SPR studies. Notwithstanding, the NKR-P1 interaction interface is very similar in primary and secondary modes; therefore, some of the previously proposed NKR-P1 interaction residues are also involved in the secondary mode (Arg181, Asp183, Tyr198, and Tyr201). Interestingly, residue LLT1:Lys169 establishes several contacts with NKR-P1 (Arg181, Ser199, and Glu200) in secondary interaction mode. However, the previously proposed pair Lys169/Glu205[39,40] is not observed in this mode either, and these residues are even farther apart – ca 11 Å. The presence of Lys169 in both primary and secondary interaction interfaces of LLT1 suggests that this residue plays an important role in the overall complex formation. Accordingly, the previously reported LLT1 mutation Lys169Glu[39,40] would lead to a co-localization of several negative side chains in the secondary interface (Glu200 and Asp183 of NKR-P1, and mutated Glu169 of LLT1), thus indicating that the disruption of the NKR-P1:LLT1 interaction observed in the previous SPR experiments most likely resulted from the weakening of the secondary rather than the primary interface. J. Kamishikiryo subsequently restored binding by introducing the Glu205Lys mutation in NKR-P1. Based on our structure, this effect would be explained by strengthening the primary interface. Thus, the previously published SPR-based interaction data largely agree with the interaction modes observed between NKR-P1 and LLT1 in the present crystal structure, although some previously suggested interaction pairs were misassigned and are not present in either interaction interface.

Several authors have previously suggested an avidity effect of multimerization upon the interaction that compensates for the low affinity of the NKR-P1:LLT1 complex[37,40,47]. Interestingly, in the present structure of this complex, we indeed observe the formation of a chain of repeating NKR-P1 and LLT1 homodimers. This pseudo-linear multimer has a zigzag shape in which the membrane-proximal parts of the two proteins are on opposite sides (Fig. 8), and it is structurally based on the alternating helix α1/α2-centered NKR-P1/LLT1 homodimers (the chain-forming effect) and the simultaneous involvement of both primary and secondary interaction modes (steric effect). On the contrary, an artificially constructed multimeric model of NKR-P1:LLT1 engaged in just the primary mode shows a chain of homodimers with a non-linear, almost helical conformation (Supplementary Fig. 6c). Furthermore, the NKR-P1 stalk regions sterically clash, and the LLT1 stalk regions are exposed outside the complex core in many different directions. Thus, such a multimer is unlikely compatible with cell

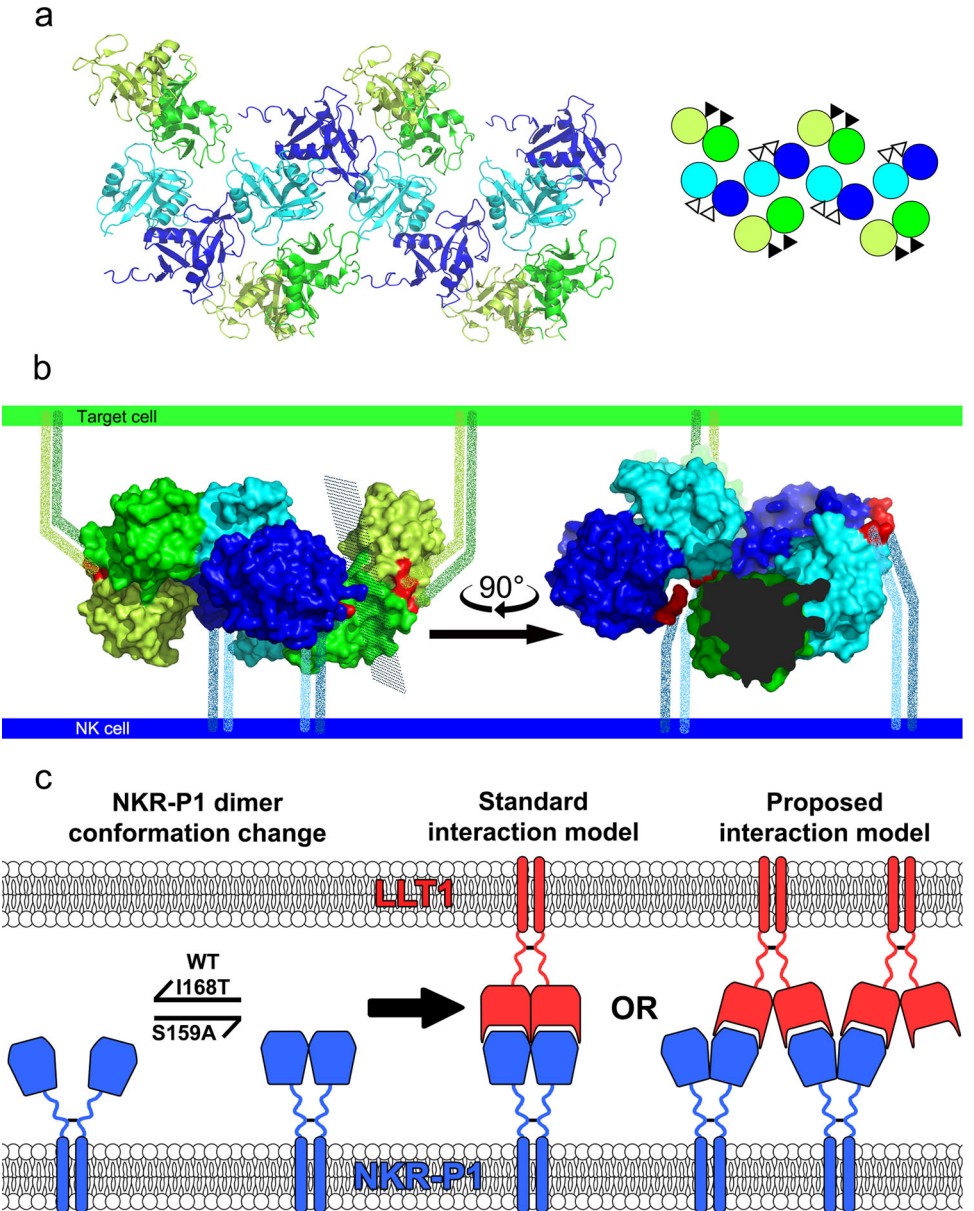

**Fig. 8 | Organization of the NKR-P1:LLT1 complexes on the cell surface.**
**a** Representation of four adjacent asymmetric units within the NKR-P1 complex crystal, excluding the additional unrelated NKR-P1 dimer. The NKR-P1 (blue and cyan) and LLT1 (green and lemon) dimers alternate in primary (cyan and green) and secondary (blue and lemon) interactions, forming a chain-like structure. The schematic depiction of this arrangement is shown in the inset with the same color code. The black and white triangles represent N-termini positions, pointing behind and in front of the display plane, respectively. **b** Depiction of the hypothetical arrangement of the chain-like structure upon contact of an NK cell (bottom, blue) with a target cell (top, green) showing the crystal structure of two NKR-P1 dimers (cyan and blue) interacting with two LLT1 dimers (green and lemon) in the primary (cyan and green) and secondary (blue and lemon) modes. The first three N-terminal residues in the structures are highlighted in red. The flexible stalk regions connecting the N-termini and cell membranes are represented as speckled lines of the corresponding color-coding. The view on the right-hand side is clipped for clarity at the plane indicated on the left-hand side view. **c** Schematic depiction of NKR-P1 extracellular domain dynamics and possible ligand binding arrangements. NKR-P1 is expressed as a disulfide-linked homodimer; however, its CTLDs may undergo conformation change similar to monomer-dimer equilibrium. Such putative equilibrium would be shifted towards monomeric species for the wild-type protein and its I168T allelic variant[51]. At the same time, a dimeric arrangement corresponding to the non-covalent dimer observed in the herein described crystal structures would be promoted for the S159A variant (left-hand side). Such NKR-P1 dimer could then interact with the cognate LLT1 ligand (itself being expressed as a disulfide-linked homodimer as well and forming stable non-covalent dimers with its CTLDs) in the previously suggested standard model of NK cell receptor – CTL ligand interaction (middle) or alternate with the dimeric ligand in the proposed chain-like arrangement based on NKR-P1:LLT1 complex crystal structure (right-hand side).

membrane anchoring and formation within the immune synapse. NKR-P1:LLT1 engaged in just the secondary mode would also form a helical multimer with both receptor and ligand stalk regions pointing in many different directions outside the complex core (Supplementary Fig. 6d). At the same time, when bound to LLT1 dimer in both primary and secondary interaction mode, the neighboring NKR-P1 molecules create a mutual accessory contact of not an insignificant energetic contribution to the stability of the whole assembly (Fig. 4a, marked with a black star). This accessory contact encompasses a similar number of hydrogen bonds and size as the secondary binding interface between NKR-P1 and LLT1 itself (Supplementary Fig. 6e), and it could thus be considered an additional stabilizing element favoring a combination of the primary and secondary binding modes rather than a single one of them.

To determine whether such zipper-like oligomers of NKR-P1:LLT1 occur not only in crystal and to some extent in solution (Fig. 5 and Supplementary Fig. 5) but also on the cell surface, we used single-molecule localization microscopy to examine full-length NKR-P1 transfectants labeled with anti-NKR-P1 AlexaFluor® 647 mAb in the presence or absence of soluble LLT1 and LLT1$^{SIM}$ (Fig. 6). We observed a significant increase in the NKR-P1 cluster of fluorescent events size and area upon adding LLT1, but not its secondary interaction mode mutant, LLT1$^{SIM}$. Although the experimental setup with one binding partner embedded in the membrane and the other being presented as soluble does not fully describe the biological reality of intercellular contact, we assume that limiting the interaction to a two-dimensional space between membranes of two adjacent cells in the case of full-length disulfidic proteins would only strengthen it. Our experiment aimed to test the hypothesis of the NKR-P1 receptor's ability to form clusters (cross-linked oligomers) in the context of the cell membrane upon engaging the dimeric species of LLT1. This experimental setup allowed us to normalize the concentration of LLT1 throughout the measurement while ensuring the presence of primarily dimeric LLT1 in the solution. Additionally, our SEC-SAXS data were best-fit with multimer chains in alternating primary and secondary interaction modes (Supplementary Fig. 5), even though the whole library of all possible models, including those based purely on primary or secondary mode, was included in the analysis. Acceptable $\chi^2$ values were obtained primarily for multimer models containing the alternating arrangement. Taken together, we conclude that combining both interaction modes is necessary for a biologically plausible multimeric interaction, as further supported by the NK cell-mediated cytotoxicity assay results (Fig. 7).

Albeit such functional multimerization of NK CTLRs has been mostly overlooked, the formation of similar nanoclusters is well described for interaction between the immunoglobulin family of KIRs and MHC class I glycoproteins[53] or interaction between KIR2DL1 and NKG2D[54]. Moreover, a periodic zipper-like network of interacting dimers was reported in the crystal structures of the co-stimulatory immunocomplexes B7-1:CTLA-4 and B7-2:CTLA-4[55,56]. Interestingly, B7-1 is expressed on the cell surface in a dynamic equilibrium between monomers and non-covalent dimers. Upon interaction with the co-stimulatory receptor CD28, B7-1 forms an interaction network composed of B7-1 and CD28 homodimers. The uncoupling of this interaction is facilitated by B7-1 dissociation to monomers, whereas insertion of B7-1 obligate dimer leads to prolonged, abnormal signaling between antigen-presenting cells and T cells[57].

Although as full-length proteins, both LLT1 and NKR-P1 form covalent disulfide homodimers on the cell surface[7,36], to our knowledge, the dimeric state of their CTL ectodomains has not yet been assessed in live cells. Conversely, the formation of the helix α2-centered non-covalent homodimer of soluble LLT1 ectodomain has been previously characterized[40,43,44] and likely occurs within the full-length protein as well. The human NKR-P1 helix α1-centered dimer uses fewer intermolecular contacts, has a smaller contact surface area than helix α2-centered dimeric CTLRs, and is thus less stable. Nevertheless, its formation would expectedly increase in the context of the full-length NKR-P1 receptor disulfide homodimer. At the same time, the length of the NKR-P1 stalk region (25 amino acids) confers enough flexibility for CTLD monomer/dimer equilibrium within the disulfide homodimer of the full-length receptor itself, which would be then further regulated by polymorphism and glycosylation heterogeneity at the NKR-P1 dimerization interface (Fig. 8c, bottom left). Thus, similarly to the B7-1:CD28 system, an equilibrium between the monomeric and dimeric states of CTL ectodomains, modifying and fine-tuning the ability of NKR-P1 to form stable higher-order complexes with LLT1, may consequently regulate the strength and signaling of the NKR-P1:LLT1 system, while cross-linking of NKR-P1 by LLT1 within the immune synapse may provide enough avidity for stable signal transduction by this low-affinity interaction complex.

In conclusion, the presented data show that the crystal structure of the NKR-P1:LLT1 complex constitutes a novel way of C-type lectin-like NK cell receptor:ligand multimerization on the cell surface that explains how ligand binding overcomes low affinity through receptor cross-linking within the immune synapse.

## Methods

### Protein expression and purification

Stabilized H176C form of the soluble LLT1 ectodomain (Gln72-Val191) was transiently expressed in HEK293S GnTI⁻ cells, as previously described[44]. The N120R, R153E, K169A secondary interaction mode mutant LLT1$^{SIM}$ was cloned and produced in the same way. The C-type lectin-like domain of human NKR-P1 was produced similarly in stably transfected HEK293S GnTI⁻ cells[50]. Briefly, the expression construct corresponding to the extracellular CTL domain of NKR-P1 (Gly90-Ser225) was subcloned into the pOPINGGTneo plasmid (kindly provided by Prof. Ray Owens, University of Oxford), flanked by the N-terminal secretion leader and by the C-terminal His-tag (with the ETG and the KHHHHHH at the N- and C-termini of the secreted protein, respectively). Suspension culture of HEK293S GnTI⁻ cells[58] was transfected with a 1:3 (w/w) mixture of the expression plasmid and 25-kDa linear polyethyleneimine. The stably transfected cell pool was selected on 200 ng/μl G418 within three weeks. The secreted proteins were purified from the harvested media by two-step chromatography – an IMAC was performed on a Talon column (GE Healthcare), followed by SEC on a Superdex 200 10/300 GL (GE Healthcare) in 10 mM HEPES pH 7.5 with 150 mM NaCl and 10 mM NaN₃. For dSTORM microscopy and in vitro NK cell assays (see below), the buffer was exchanged for a tissue culture grade PBS (137 mM NaCl, 2.7 mM KCl, 10 mM Na₂HPO₄, 1.8 mM KH₂PO₄, pH 7.4). For deglycosylation, GST-fused Endo F1[59] was added in a 1:100 weight ratio to proteins in SEC buffer with 50 mM citrate pH 5.5 and incubated for 2 h at 37 °C. The deglycosylated proteins were then purified by batch affinity chromatography on Glutathione Sepharose 4B resin (GE Healthcare) followed by SEC, as described above.

### Crystallization

*NKR-P1 glycosylated (structure NKR-P1_glyco)* – Soluble human NKR-P1 ectodomain at 20 mg/ml in SEC buffer was crystallized using the sitting drop vapor diffusion method. Drops (100 nl of reservoir solution and 100 nl of protein solution) were set up using a Cartesian Honeybee 961 robot (Genomic Solutions) at 294 K. The reservoir consisted of 20% (w/v) PEG 3350, 200 mM di-sodium tartrate pH 7.2 (PEG/Ion screen, condition 36; Hampton Research). A hexagonal crystal with dimensions of 150 × 150 × 20 μm was cryoprotected by soaking in the reservoir solution with the addition of 25% (v/v) ethylene glycol.

*NKR-P1 deglycosylated (structure NKR-P1_deglyco)* – The Endo F1-deglycosylated soluble human NKR-P1 ectodomain was concentrated to 12 mg/ml and crystallized as described above. The reservoir consisted of 20% (w/v) PEG 3350, 200 mM ammonium fluoride, and 200 mM lithium chloride pH 6.2 (PEG/Ion screen, condition 3, Additive screen, condition 17; Hampton Research). A 50 × 50 × 150 μm rod-shaped crystal was cryoprotected as above by adding 25% (v/v) glycerol.

*NKR-P1:LLT1 complex (structure NKR-P1:LLT1)* – The Endo F1-deglycosylated soluble human NKR-P1 and LLT1 ectodomains were mixed at a 1:1 molar ratio and concentrated to 8 mg/ml of total protein concentration. The protein complex was crystallized as described above; drops (200 nl of the reservoir and 100 nl of protein solutions) were seeded with 50 nl of stock solution of crushed needle-shaped crystals of deglycosylated NKR-P1 grown in 20% (w/v) PEG 3350, 200 mM ammonium fluoride pH 6.2 (PEG/Ion screen, condition 3; Hampton Research). The reservoir consisted of 200 mM ammonium sulfate, 20% (w/v) PEG MME 5000, 100 mM Tris pH 7.5 (Proplex screen, condition 1-40; Molecular Dimensions). A tetragonal bipyramid crystal of dimensions 30 × 30 × 80 μm was cryoprotected as detailed above by adding 25% (v/v) glycerol.

## X-ray diffraction data collection, processing, and structure solution

All diffraction data were collected at the Diamond Light Source (Harwell, UK) at beamline I03 using a wavelength of 0.97625 Å and a PILATUS3 6 M detector. The crystal-detector distance was set to 340 mm, exposure time per image was 0.02 s, oscillation width was 0.1°, and the temperature was 100 K. 7200 images were collected for each dataset. In the case of the NKR-P1:LLT1 complex, only 5000 images were finally used for data processing. All diffraction images were indexed and integrated using the XDS package[60], scaled using AIMLESS[61], and 5% of randomly selected reflections were utilized as an $R_{free}$ set. The phase problem was solved by molecular replacement – NKR-P1_glyco: in program BALBES[62] using the structure of the human NK cell receptor KLRG1 bound to E-cadherin (https://doi.org/10.2210/pdb3FF7/pdb)[63]; NKR-P1_deglyco: 6 chains found in PHASER[64] using murine NKR-P1A (https://doi.org/10.2210/pdb3T3A/pdb)[65] were completed with 2 chains found in MOLREP[66]; NKR-P1:LLT1: 4 chains found in BALBES as NKR-P1 chains (using the structure of murine dectin-1, https://doi.org/10.2210/pdb2BPD/pdb)[48] were completed with two more chains in MOLREP, and all 6 chains were manually reinterpreted as 4 NKR-P1 chains and as 2 LLT1 chains. Refinement was performed using REFMAC5[67] and manual editing in COOT[68]. The last cycle of refinement was completed using all reflections. The final data processing and structure parameters are outlined in Table 1.

## Crystal structure quality assessment

*NKR-P1_glyco* – The structure, comprising one dimer of the glycosylated human NKR-P1 CTLD, is overall well defined in the electron density map, corresponding to the high resolution of the structure (1.8 Å). Glycosylation at the dimerization interface does not show the overlapping features observed in the structures of deglycosylated NKR-P1 (below). GlcNAc was modeled on Asn116 with full occupancy in both chains, whereas glycosylation at Asn157 was not observed in the electron density map. All modeled glycosylation chains (GlcNAc$_2$Man$_5$ at A/Asn169, GlcNAc$_2$Man$_3$ at B/Asn169, and GlcNAc at residues Asn116 in both chains) are well localized in the electron density map.

*NKR-P1_deglyco* – The asymmetric unit comprises four dimers of the human NKR-P1 CTLD deglycosylated after the first GlcNAc unit. The length of the localized part of the protein chain varies from the shortest chains A, F, and G, with modeled residues Leu91–Leu214, to the longest chain H, with residues Gly90–Arg218. GlcNAc at residue Asn169 is well localized, while GlcNAc units bound to Asn116 and Asn157 at the dimerization interface are present in alternative and overlapping positions; residue Asn116 also shows alternative conformers. GlcNAc units attached to Asn157 and Asn116 were modeled with 0.5 occupancies, and only the most distinct units from each overlapping pair were modeled (Table 1).

*NKR-P1:LLT1* – The asymmetric unit contains two dimers of the human NKR-P1 CTLD and one dimer of the LLT1 CTLD. The structure has a well-defined electron density map, and all protein chains can be unambiguously assigned. The most distinct difference peaks correspond to non-interpretable small ligands. LLT1 has well-localized GlcNAc units at residues Asn95 and Asn147. Localized GlcNAc units of NKR-P1 are the same as those identified in the NKR-P1_deglyco structure (previous paragraph).

## CD spectroscopy

Circular dichroism (CD) spectra of wild-type and S159A NKR-P1 and wild-type and SIM LLT1 were recorded using a Chirascan Plus CD spectropolarimeter with Pro-Data Chirascan software (Applied Photophysics) and a 0.1 cm pathlength quartz cell. Spectra were recorded over the wavelength range of 195–260 nm in steps of 1 nm at room temperature. The sample concentrations were 0.2 mg/ml and 0.3 mg/ml for NKR-P1 and LLT1 protein samples, respectively, in 10 mM HEPES, 150 mM NaCl pH 7.5 sample buffer. The CD signal was expressed as ellipticity, and the resulting spectra were buffer subtracted. Secondary structure composition was analyzed using the CDNN 2.1 software (Applied Photophysics).

## Analytical ultracentrifugation

NKR-P1:LLT1 complex formation and dimerization of NKR-P1 S159A mutant were analyzed in an analytical ultracentrifuge ProteomeLab XL-I (Beckman Coulter)[69]. For the sedimentation velocity experiment, samples of glycosylated NKR-P1, LLT1, and their equimolar mixtures with increasing concentrations in SEC buffer were spun in the An-50 Ti rotor (Beckman Coulter) at 48,000 rpm at 20 °C, and 150 scans with 0.003 cm spatial resolution were recorded in 5-min steps using absorbance optics. The centrifuge was operated using ProteomeLab software (Beckman Coulter). Buffer density and protein partial specific volumes were estimated in SEDNTERP (http://www.jphilo.mailway.com). Data were analyzed with SEDFIT[70] using the continuous c(s) distribution model. Binding isotherms (and figures illustrating AUC data) were prepared in GUSSI[71] and then best-fit in SEDPHAT[72] using hetero-association models A + B ⇔ AB or A + B ⇔ AB + B ⇔ ABB, where A is the LLT1 dimer and B is NKR-P1 monomer, respectively. Only $K_D$ and sedimentation coefficients of AB or ABB were floated in the fit; the other parameters were kept constant at known values.

## Microscale thermophoresis

For the microscale thermophoresis (MST) measurements of NKR-P1:LLT1 and NKR-P1:LLT1$^{SIM}$ interactions, the NKR-P1 was fluorescently labeled with Atto 488 NHS-ester (Merck) at pH 6.5; the excess label was removed by size exclusion chromatography. Fluorescently labeled 100 nM NKR-P1 was mixed with dilution series of LLT1 or LLT1$^{SIM}$ and analyzed in standard capillaries in an NT.115 Monolith using the MO.Control software (NanoTemper), 30% LED excitation power, and 60% MST power. Raw data were analyzed in PALMIST[73]; the exported isotherms were best-fit in SEDPHAT[72] and figures prepared in GUSSI[71].

## Small-angle X-ray scattering

SEC-SAXS data for the NKR-P1:LLT1 complex were collected at the Diamond Light Source (Didcot, UK) at beamline 21 using an Agilent 1200 HPLC system with 2.4 mL Superdex 200 column (GE Healthcare), a Pilatus P3-2M detector, 12.4 keV radiation, and 4.014 m sample-to-detector distance. The human NKR-P1 and LLT1 ectodomains with GlcNAc$_2$Man$_5$ glycans diluted in 10 mM HEPES, 150 mM NaCl, 10 mM NaN$_3$, pH 7.5 were mixed at a 1:1 molar ratio. The data were collected at 293 K for buffer and protein samples at a 15 mg/ml loading concentration. The data in selected intervals (frames 355–367, 368–378, 379–388, 388–399, 431–440, 481–491) were solvent-subtracted in SCÅTTER (developed by Robert Rambo at the Diamond Light Source, https://www.bioisis.net/tutorials/9) and then in the intervals separately merged and characterized using the ATSAS[74] package. As proof of the data quality, the scattering plot, Kratky plot, and pair distance distribution function P(r) are shown in Fig. 5 for intervals 389–399 (a sample data range from the first peak) and 481–491 (a sample data range from the second peak). The scattering and Guinier plots for all the data ranges are shown in Supplementary Fig. 5. Agreement between the SAXS data and our 3D crystal structure of the NKR-P1:LLT1 complex was evaluated using OLIGOMER[75]. Library of NKR-P1 and LLT1 structural models representing monomers, dimers, and their interacting multimers in various permutations of primary and secondary interaction modes was generated in PyMOL from the herein solved crystal structures by applying symmetry operations (in total, 49 different models). The OLIGOMER algorithm was then left to select their combinations to best-fit all individual scattering curves and curves resulting from the six selected merged data intervals (Supplementary Data 1 and Supplementary Fig. 5).

## Super-resolution dSTORM microscopy

Full-length NKR-P1 stable transfectants were generated in HEK293S GnTI⁻ cell line using the piggyBac transposon-based system with doxycycline-inducible protein expression[76]. Microscopy samples were prepared from cells treated with 5 ng/ml doxycycline overnight. Cells were washed with PBS and incubated for 1 h at 37 °C with 8.4 mg/ml LLT1 or LLT1[SIM] in PBS or PBS alone. After incubation, cells were allowed to settle on the surface of PLL-coated glass slides for 25 min at 37 °C. Cells were then fixed with 4% PFA and 0.2% GA for 10 min at room temperature and washed three times with PBS. For antibody staining, cells were first blocked with 5% BSA for 30 min and stained with Alexa Fluor® 647 anti-human CD161 antibody (clone HP-3G10; BioLegend) at 10 μg/ml in blocking solution for 60 min. Samples were washed five times with PBS before post-fixation with 4% PFA and 0.2% GA for 10 min. Finally, cells were treated with 15 mM $NH_4Cl$ and washed twice with PBS. dSTORM images were acquired with Elyra PS.1 using Zen Black edition software (Carl Zeiss). Fiducial markers (FluoSpheres F8801; Thermo Fisher) were diluted in PBS and allowed to settle on the sample for 1 h (the final dilution of the bead stock was 100,000×). Before each measurement, the buffer was exchanged for glucose oxidase/catalase/MEA-based imaging buffer, and the sample was sealed with cover glass and silicon to prevent buffer oxidation. For each cell, $2 \times 10^4$ raw images were acquired in HP TIRF illumination mode with an exposure time of 15 ms, using 100% of 642 nm laser power and 100×, 1.46 numerical aperture, oil immersion objective. We used the same batch and concentration of the antibody to ensure consistent labeling within the single measurements for all experiments. Moreover, we always measured conditions with and without the soluble ligand in each series of individual measurements to ensure consistent overall data quality across the whole study.

## Super-resolution image data analysis

Super-resolution dSTORM images were reconstructed from raw image sequences with the ThunderSTORM plug-in[77] for the ImageJ processing software[78]. Sub-pixel localization of molecules was performed by fitting an integrated Gaussian point-spread function models using the maximum likelihood estimation fitting method[79]. Reconstructed dSTORM images were corrected for drift using fiducial markers and for multiple localizations from a single source by merging events within 20 nm appearing in subsequent frames (with an off-gap of 5 frames). Voronoi tessellation cluster analysis was performed in ClusterViSu[80] on a whole inner cell surface (cell edges and other concentrated membrane regions were excluded from analysis). A threshold value of 250 was chosen by comparing the distribution of Voronoi polygon surface values between localizations in the experimental areas and randomized regions of the same density of events. Clusters containing less than two fluorescent events were discarded from the data set. Multiple means were compared with one-way analysis of variance (ANOVA) with Bonferroni correction in OriginPro 2018. Graphs show mean values, and error bars represent the S.D. Values of $p > 0.05$ are indicated as not significant; statistically significant $p$ values are indicated with asterisks (*$p < 0.05$, **$p < 0.01$, ***$p < 0.001$, ****$p < 0.0001$).

## NK cell-mediated cytotoxicity assay

The influence of LLT1 or LLT1[SIM] on inhibition of the cytotoxic activity of primary NK cells was assessed by flow cytometry. Buffy coats for human NK cell isolation were purchased from the Institute of Hematology and Blood Transfusion (IHBT, Prague, Czech Republic) as material for research use. The IHBT arranged the consent of the donors. Primary NK cells were isolated by negative selection using an NK cell isolation kit (Miltenyi Biotec) according to the manufacturer's protocol. Purified NK cells were cultured overnight at $1 \times 10^6$ cells/ml in RPMI 1640 supplemented with 10% FCS, 100 units/ml penicillin, 100 μg/ml streptomycin, and 80 ng/ml IL-2 (Sigma–Aldrich) for their activation. K562 target cells were stained with CellTrace Violet

Proliferation Kit (Thermo Fisher). Following the staining, $1 \times 10^4$ target cells were mixed with activated NK cells in a 40:1 (E:T) ratio, and LLT1 or LLT1[SIM] proteins were added, keeping the final reaction volume at 20 μl (6.5 μl of K562 and 6.5 μl of NK cell suspensions and 7 μl of the concentrated protein stock solutions in PBS, to a final protein concentration of 50 or 250 μM). After 4 h of incubation, the cell mixture was centrifuged at $300 \times g$ and stained with 1 μg/ml 7-AAD. Cells were analyzed using a BD LSR II flow cytometer and BD FACSDiva software (BD Biosciences) in three independent cytotoxicity assays performed using NK cells from three different healthy donors in triplicates. Data were analyzed in FlowJo software (BD Biosciences). The gating strategy for flow cytometry analysis is shown in Supplementary Fig. 7. Data are presented as a mean of measurements with S.D. When applicable, data were statistically evaluated by one-way ANOVA with values of $p < 0.05$ considered statistically significant.

## Reporting summary

Further information on research design is available in the Nature Research Reporting Summary linked to this article.

## Data availability

The refined coordinate and structure factor files for the X-ray crystal structures reported in this study have been validated by the Protein Data Bank (https://www.wwpdb.org) and deposited there under the accession numbers of https://doi.org/10.2210/pdb5MGR/pdb (NKR-P1_glyco), https://doi.org/10.2210/pdb5MGS/pdb (NKR-P1_deglyco), and https://doi.org/10.2210/pdb5MGT/pdb (NKR-P1:LLT1). Links to the other PDB entries in this paper are https://doi.org/10.2210/pdb2BPD/pdb, https://doi.org/10.2210/pdb2CL8/pdb, https://doi.org/10.2210/pdb3FF7/pdb, https://doi.org/10.2210/pdb3T3A/pdb, https://doi.org/10.2210/pdb4IOP/pdb, https://doi.org/10.2210/pdb4QKI/pdb, https://doi.org/10.2210/pdb5J2S/pdb, and https://doi.org/10.2210/pdb6E7D/pdb. Diffraction data have been deposited in the SBGrid Data Bank under the codes 778 (NKR-P1_glyco; https://doi.org/10.15785/SBGRID/778); 779 (NKR-P1_deglyco, https://doi.org/10.15785/SBGRID/779) and 780 (NKR-P1:LLT1; https://doi.org/10.15785/SBGRID/780). Diffraction data from the SEC-SAXS experiment have been deposited to Mendeley Data (https://doi.org/10.17632/268ww2m4j31)[81]. The total output of the OLIGOMER analysis of the SEC-SAXS data is available as Supplementary Data 1. Source data are provided with this paper.

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

## Acknowledgements

This study was supported by the Czech Science Foundation grants 15-15181 S and 18-10687 S to O.V., the Ministry of Education, Youth and Sports of the Czech Republic grant LTC17065 to O.V. (in the frame of the COST Action CA15126 MOBIEU), the Charles University Grant Agency projects 161216 and 1378219 to J.B. and B.K, and the European Regional Development Fund (CZ.02.1.01/0.0/0.0/15_003/0000447). Microscopy was performed in the Laboratory of Confocal and Fluorescence Microscopy, co-financed by the European Regional Development Fund and the state budget of the Czech Republic (projects no. CZ.1.05/4.1.00/16.0347 and CZ.2.16/3.1.00/21515) and supported by the Czech-BioImaging large RI project LM2018129. Computational resources were supplied by the project "e-Infrastruktura CZ" (e-INFRA LM2018140) provided within the program Projects of Large Research, Development, and Innovations Infrastructures. We acknowledge the use of CF Biophysical methods of CMS, CIISB, Instruct-CZ Centre, supported by MEYS CR (LM2018127). The authors wish to thank Dr. Carlos V. Melo for critically proofreading the manuscript. The authors also acknowledge the support and the use of resources of Instruct-ERIC through the R&D pilot scheme APPID 56 and 286 to O.V. and J.B., and of the BioStruct-X EC FP7 project 283570. The Wellcome Centre for Human Genetics is supported by the Wellcome Trust (grant 090532/Z/09/Z). We thank

Diamond Light Source for beamtime (proposal MX10627) and the staff of beamlines I03 and 21 for assistance with data collection.

## Author contributions

J.B., O.S., B.K., S.P., E.P., and C.A. contributed to protein expression and purification; J.B. and Y.Z. performed the protein crystallization; Y.Z. and K.H. performed the X-ray diffraction measurements; J.B., T.S., J.S., T.K., J.Du., and J.Do. contributed to the data processing and model refinement; O.S. performed the SEC-SAXS data measurement; J.B. and T.S. performed the SAXS data analysis; O.V. performed the analytical ultracentrifugation measurements and analysis; J.B. and B.K. acquired and analyzed the dSTORM data; B.K., V.G., and D.C. performed the NK cytotoxicity assay; J.B., T.S., J.Do., and O.V. designed the experiments and wrote the manuscript with critical input from J.H.

## Competing interests

The authors declare no competing interests.
