## [Peer Review File · Nature Communications]

Structure of the human NK cell NKR-P1:LLT1 receptor:ligand complex reveals clustering in the immune synapseREVIEWER COMMENTS

Reviewer #1 (Remarks to the Author):

In the revision of the manuscript initially submitted about 4 years ago, the authors have constructively addressed the main concerns of the reviewer. Indeed, the authors are to be congratulated in substantiating the main conclusion of the paper by conducting a series of complementary solution based and single molecule imaging experiments while weaving in some analyses of the mouse NKR-P1b complex structures. In doing so the authors present a compelling revision that provides new mechanistic insight into human NKR-P1- LLT1 structure and oligomerisation and how this relates to an increased understanding of NK cell function.

Reviewer #2 (Remarks to the Author):

My review relates to the SEC-SAXS analysis that was carried on after the first round of reviews. As stated by Reviewer 1, "in the NKR-P1/LLT1 crystal, each monomer of the LLT1 homodimer is observed to contact a different monomer of the NKR-P1 homodimer. The primary interaction mode corresponds closely to the interface found in the NKp65/KACL complex,..., but evidence for the biological relevance of the secondary interaction mode is much less convincing." SEC-SAXS was then a sound approach to "confirm" the existence of such an interaction in solution. Unfortunately, the way the SEC-SAXS experiments were carried on and analysed could not answer the question. Since the complex NKR-P1/LLT1 is in equilibrium with the individual partners, it necessarily dissociates during its elution through the SEC column, even if the concentration of the injected solution is high. It is therefore no surprise that the main elution peak corresponds to several species eluting closely to each other. This reflects well in the R_g profile of Figure 4, for which no clear plateau is seen. The authors then choose a strange strategy, consisting in merging the curves from several peaks (even those from higher oligomeric species) and analyse them as a large ensemble. It seems to me that this strategy, by blurring back the SEC-SAXS information, contradicts the very interest of SEC-SAXS, which is to isolate the individual curves, and analyse them on their own. The conclusions drawn by the analysis based on OLIGOMER, are necessarily more speculative than what the analysis of a curve from a monodisperse species would be. I would rather suggest to use a deconvolution strategy of the main elution peak, based on EFA or on US-Somo approaches. There will then be a chance to get the curves of the individual species and analyse them accordingly. Alternatively, a more experimental strategy would be to saturate the elution buffer with one of the partners during the SEC-SAXS elution, in order to shift the equilibrium towards the formation of the complex. This strategy is of course much more expensive, but it could then unambiguously provide the scattering curve of the complex alone: see for instance <https://doi.org/10.1038/emboj.2011.461>.

Reviewer #3 (Remarks to the Author):

This review is only focused on the super-resolution imaging portion of the paper.

The authors present data demonstrating that NKR-P1 increases in average cluster size and events per cluster in the presence of soluble LLT1, and that this change in distribution requires the presence of the three predicted strongest contacts in the LLT1 secondary interaction interface. While the data is consistent with the other findings with complementary methods, the manuscript would be strengthened by further justification of several experimental and analysis design choices and the inclusion of other relevant controls to ensure consistent labeling of the proteins imaged.

One comment is regarding the experimental design, in which LLT1 and LLT1-SIM are introduced to the cells in a soluble form, which differs dramatically from the presentation of LLT1 on the surface of an activated monocyte, B cell, Nk or T cell. It would be relevant for the authors to address how this change in presentation of LLT1, and thus the biophysical constraints that would be placed upon LLT1 if it were attached to a cellular membrane, may affect its binding to NKR-P1.

Also, because dSTORM imaging can be very affected by the number of labels on the secondary

antibody, it would be useful to determine how the author's determined that the NKR-P1 homodimers were doubly labeled with Alexa-647. And additionally, the determination of the 15 nm localization precision, as this can be greatly affected by dyes per secondary antibody, packing of multiple dyes in a small area leading to either quenching or fret effects on the emission, under labeling of the target, and labeling of the target with secondary antibodies with no dye or bleached dyes conjugated. Since the change in size and event number was modest, it is relevant to report how the authors determined that a majority of NKR-P1 molecules were being labeled with single dyes. This is especially the case as most commercial secondary dyes are labeled with 4-7 dyes per antibody. Because the changes in size and events observed fall within the StDev and the localization precision, it would be relevant to provide additional controls to support the significance of this change. And to ensure that the statistical tests between the measurements account for multiple measurements. Several distributions appear to have more than one population present, possibly single homodimers and multimers, and in that case reporting the mean value may not be as relevant as changes between the overall histograms of the distributions, which do not preassume a single population.

Further support of multimerization at this level could be provided by observing single or up to 3-step permanent photobleaching of antibodies in the absence of the MEA buffers to promote blinking, or with another secondary fluorophore that is more stable.

Also- it may be relevant to discuss whether a single homodimer is considered a cluster? or if another terminology may be more relevant to distinguish between lightly cross-linked homodimers, versus singly-distributed homodimers.

REPLY TO REVIEWERS' COMMENTS

Reviewer #1 (Remarks to the Author):

In the revision of the manuscript initially submitted about 4 years ago, the authors have constructively addressed the main concerns of the reviewer. Indeed, the authors are to be congratulated in substantiating the main conclusion of the paper by conducting a series of complementary solution-based and single-molecule imaging experiments while weaving in some analyses of the mouse NKR-P1B complex structures. In doing so, the authors present a compelling revision that provides new mechanistic insight into human NKR-P1-LLT1 structure and oligomerisation and how this relates to an increased understanding of NK cell function.

We are grateful to the reviewer for the positive feedback and hope we will continue contributing to understanding NK cell function in the future.

Reviewer #2 (Remarks to the Author):

My review relates to the SEC-SAXS analysis that was carried on after the first round of reviews.

As stated by Reviewer 1, "in the NKR-P1/LLT1 crystal, each monomer of the LLT1 homodimer is observed to contact a different monomer of the NKR-P1 homodimer. The primary interaction mode corresponds closely to the interface found in the NKp65/KACL complex, but evidence for the biological relevance of the secondary interaction mode is much less convincing." SEC-SAXS was then a sound approach to "confirm" the existence of such an interaction in solution.

We thank the reviewer for commenting on the SEC-SAXS part of our research. Indeed, at the time of first submitting this manuscript, the presence of the secondary binding mode in solution was assessed by the SAXS experiment performed in batch mode, and we were not able to deconvolute the data corresponding to a mixture of monomers, dimers, and higher interacting species. This is indeed why we have chosen to proceed with the SEC-SAXS measurement, thus separating the higher oligomeric species from dissociating monomers/dimers. Nevertheless, we would like to point out that since the first submission, we have complemented the evidence for the presented secondary binding mode also with a nanoscopy experiment assessing the formation of clusters on cellular surface and by observing the direct effect on NK cell inhibition. Thus, the SEC-SAXS data presented here are not the only experimental evidence for the biological relevance of the secondary binding mode observed in the crystal structure of the NKR-P1/LLT1 complex.

Unfortunately, the way the SEC-SAXS experiments were carried on and analysed could not answer the question. Since the complex NKR-P1/LLT1 is in equilibrium with the individual partners, it necessarily dissociates during its elution through the SEC column, even if the concentration of the injected solution is high. It is therefore no surprise that the main elution peak corresponds to several species eluting closely to each other. This reflects well in the R_g profile of Figure 4, for which no clear plateau is seen. The authors then choose a strange strategy, consisting in merging the curves from several peaks (even those from higher oligomeric species) and analyse them as a large ensemble. It seems to me that this strategy, by blurring back the SEC-SAXS information, contradicts the very interest of SEC-SAXS, which is to isolate the individual curves, and analyse them on their own. The conclusions

drawn by the analysis based on OLIGOMER, are necessarily more speculative than what the analysis of a curve from a monodisperse species would be.

Here we have to agree with the reviewer that the analysis performed by OLIGOMER would be indeed more speculative if it were indeed based on merging curves from several resolved peaks while analyzing them as a large ensemble. However, that is not what we present in the manuscript.

As seen in Figure 4, we obtained two main resolved peaks in size-exclusion chromatography, one corresponding to the higher oligomeric species and the other containing dissociating monomeric/dimeric species. Although the Rg profile does not have an apparent plateau, we have selected four separate intervals from the first peak and two intervals from the second peak where the Rg delta environment has only a low difference. These intervals are denoted in Figure 4 as columns with diagonal hatching with corresponding frame numbers in the figure legend. Data from these intervals were then merged separately – we have not mixed or merged data between these intervals. We have performed OLIGOMER analysis on single curves throughout the data as well (this is now shown in the Supplementary Data in the revised manuscript); however, due to the low signal of the first peak that contains the more interesting oligomeric species, it proved necessary to boost the signal-to-noise ratio by merging small intervals with low local Rg difference in order to fit the imperfectly resolved mixture of oligomeric species within the data. We agree this approach does not entirely deconvolute the data; however, it proves the point that we are observing larger oligomeric species in solution which are in equilibrium with monomer/dimeric species, and it also shows that the oligomeric models based on our crystal structure fit the obtained SAXS curves quite well (as shown in Supplementary Figure 6).

We have changed the text in paragraph "SEC-SAXS analysis confirms NKR-P1:LLT1 higher-order complex formation" as well as in figure legends and methods to make it more transparent that we have not mixed and merged data from two distinct elution peaks.

I would rather suggest to use a deconvolution strategy of the main elution peak, based on EFA or on US-SOMO approaches. There will then be a chance to get the curves of the individual species and analyse them accordingly. Alternatively, a more experimental strategy would be to saturate the elution buffer with one of the partners during the SEC-SAXS elution, in order to shift the equilibrium towards the formation of the complex. This strategy is of course much more expensive, but it could then unambiguously provide the scattering curve of the complex alone: see for instance <https://doi.org/10.1038/emboj.2011.461>.

As per the reviewer's suggestion, we have tried to analyze the data with US-SOMO; however, this approach proved unsuccessful. Given the complexity of the mixture, i.e., monomeric and dimeric forms of both proteins interacting in the primary or secondary or both interaction modes, it is impossible to specify how exactly the peak should be deconvoluted, and we are afraid that attempting this could provide biased results. On the contrary, the analysis performed by the OLIGOMER where all possible combinations of all species were subjected as a library to the algorithm, and then the algorithm used it to best-fit the data, thus selecting the most probable combinations of species present in the samples, is completely unbiased. As mentioned above, we sampled through the SEC-SAXS data by analyzing several discrete, distinct data intervals, which at least partially deconvolutes the data, just in a different way. Even if the buffer were saturated with one of the binding partners, still, there would be an equilibrium of different oligomeric species utilizing varying ratios of the primary and secondary interaction modes, thus necessitating the same type of analysis as already applied. We believe the main confusion here was that the reviewer misunderstood how we

sampled and merged the data, and we apologize for not being clearer. The corresponding parts of the text were edited to provide the reader with a more descriptive explanation.

Reviewer #3 (Remarks to the Author):

This review is only focused on the super-resolution imaging portion of the paper.

The authors present data demonstrating that NKR-P1 increases in average cluster size and events per cluster in the presence of soluble LLT1, and that this change in distribution requires the presence of the three predicted strongest contacts in the LLT1 secondary interaction interface. While the data is consistent with the other findings with complementary methods, the manuscript would be strengthened by further justification of several experimental and analysis design choices and the inclusion of other relevant controls to ensure consistent labeling of the proteins imaged.

One comment is regarding the experimental design, in which LLT1 and LLT1-SIM are introduced to the cells in a soluble form, which differs dramatically from the presentation of LLT1 on the surface of an activated monocyte, B cell, NK or T cell. It would be relevant for the authors to address how this change in presentation of LLT1, and thus the biophysical constraints that would be placed upon LLT1 if it were attached to a cellular membrane, may affect its binding to NKR-P1.

We thank the reviewer for raising this point. Indeed, the experimental setup with one binding partner embedded in the membrane and the other being presented as soluble does not describe the biological reality of intercellular contact. However, this dSTORM experiment aimed to test the hypothesis of the NKR-P1 receptor's ability to form clusters (cross-linked oligomers) in the context of the cell membrane upon engaging the dimeric species of LLT1. This experimental setup allowed us to normalize the concentration of LLT1 throughout the measurement while ensuring the presence of primarily dimeric LLT1 in the solution. Compared to utilizing LLT1 cell transfectants, we have simplified the experimental setup to achieve higher reproducibility of the measurements. Furthermore, when we tested presenting LLT1 on supported lipid bilayers (not shown in the current manuscript), thus getting closer to the biological reality, sadly, mainly monomeric species of LLT1 were present on the lipidic bilayers at the low concentrations of LLT1 necessary for single molecule tracking and bleaching. Thus, this approach proved not suitable at the moment; however, we agree with the reviewer and have updated the manuscript's text to address the difference in LLT1 presentation.

Also, because dSTORM imaging can be very affected by the number of labels on the secondary antibody, it would be useful to determine how the authors determined that the NKR-P1 homodimers were doubly labeled with Alexa-647. And additionally, the determination of the 15 nm localization precision, as this can be greatly affected by dyes per secondary antibody, packing of multiple dyes in a small area leading to either quenching or fret effects on the emission, under labeling of the target, and labeling of the target with secondary antibodies with no dye or bleached dyes conjugated. Since the change in size and event number was modest, it is relevant to report how the authors determined that a majority of NKR-P1 molecules were being labeled with single dyes. This is especially the case as most commercial secondary dyes are labeled with 4-7 dyes per antibody. Because the changes in size and events observed fall within the StDev and the localization precision, it would be relevant to provide additional controls to support the significance of this change. And to ensure that the statistical tests between the measurements account for multiple measurements. Several distributions appear to have more than one population present,

possibly single homodimers and multimers, and in that case reporting the mean value may not be as relevant as changes between the overall histograms of the distributions, which do not presume a single population.

Further support of multimerization at this level could be provided by observing single or up to 3-step permanent photobleaching of antibodies in the absence of the MEA buffers to promote blinking, or with another secondary fluorophore that is more stable.

We thank the reviewer for this technical note. However, we used only primary antibodies in our study, directly labeled by the manufacturer with AF647. We used the same batch and concentration of the antibody to ensure consistent labeling within the single measurements for all experiments. Moreover, we always measured conditions with and without the soluble ligand in each series of individual measurements. The negative controls were also evaluated (the condition with NKR-P1 and PBS added instead of protein ligands) to ensure consistent overall measurement quality and avoid possible errors resulting from different microscope settings or sample handling.

Being aware that this experimental approach does not directly describe the size of the receptor clusters (if any) due to potential overcounting artifacts, we instead focused only on the comparison of three distinct states of the receptor: free receptor on the membrane, receptor interacting with ligand potentially inducing clustering/oligomerization by its two interaction modes, and receptor interacting with ligand bearing only primary interaction interface, and assessing the relative change of the observed cluster of events sizes between these three states.

As the cluster of events parameters are averaged over the whole cell (i.e., single plotted point corresponds to cluster parameters observed in a single cell), the statistical tests between the plotted data indeed account for multiple measurements – i.e., cluster data collected on various cells in multiple experiments.

The cluster of events size histogram, which may point at the combination of different oligomer contributions, was assessed for every single cell, but as we are not sure about the value (size) corresponding to the NKR-P1 homodimer alone (random or some pre-existing oligomers may form on the cell surface even without ligand), we were not able to deconvolute the curve.

Also - it may be relevant to discuss whether a single homodimer is considered a cluster? Or if another terminology may be more relevant to distinguish between lightly cross-linked homodimers, versus singly-distributed homodimers.

Thank you for this excellent point. Yes, we are talking about "cluster of events" parameters in the evaluation, which means a cluster of acquired fluorescent events. The size of the "cluster of events" is not the same as the "receptor cluster size", which we are not defining (due to method limitations, as discussed above). Most of these "clusters of events" (CoE) correspond to NKR-P1 homodimers or the homodimers cross-linked by the soluble dimeric ligand. Both the text in the Results and Figure legends sections has been modified to clarify this misunderstanding. Moreover, we have edited Figure 5 and substituted "cluster" for "cluster of events" (CoE) and added an explanation to its legend to avoid any confusion.

REVIEWERS' COMMENTS

Reviewer #1 (Remarks to the Author):

No further comments

Reviewer #2 (Remarks to the Author):

I thank the authors for clarifying the way the SEC-SAXS data were pooled together before using the OLIGOMER algorithm. I also understand that deconvolution of the data would reveal impossible in their case. I would then support publication.

Reviewer #3 (Remarks to the Author):

The reasoning behind the response to review and the edited text, help to clarify the conclusions and methodology and resolves the majority of my requests for clarification and any concerns regarding interpretation of results or experimental design.

The only additional detail that I would request be considered for additional prior to final publication are for more description (e.g. catalog number, name) regarding the type and concentration of the fiducials used to be added to the Methods section. And for a note in the methods section and/or document text to indicate whether edges were included in the 'whole cell' analysis, as inclusion of edge regions can cause there to be areas that are unable to measure clustering (no cell present) and apparent large clusters at cell edges where membrane is concentrated in the TIRF field. A comment regarding how edges were exclude from analysis would satisfy this concern.

REPLY TO REVIEWERS' COMMENTS

Reviewer #2 (Remarks to the Author):

I thank the authors for clarifying the way the SEC-SAXS data were pooled together before using the OLIGOMER algorithm. I also understand that deconvolution of the data would reveal impossible in their case. I would then support publication.

We thank the reviewer for supporting our publication.

Reviewer #3 (Remarks to the Author):

The reasoning behind the response to review and the edited text help to clarify the conclusions and methodology and resolves the majority of my requests for clarification and any concerns regarding interpretation of results or experimental design.

The only additional detail that I would request be considered for additional prior to final publication are for more description (e.g., catalog number, name) regarding the type and concentration of the fiducials used to be added to the Methods section. And for a note in the methods section and/or document text to indicate whether edges were included in the 'whole cell' analysis, as inclusion of edge regions can cause there to be areas that are unable to measure clustering (no cell present) and apparent large clusters at cell edges where membrane is concentrated in the TIRF field. A comment regarding how edges were excluded from analysis would satisfy this concern.

We are glad that our responses met the reviewer's remarks.

Concerning the fiducial markers, we used commercial FluoSpheres™ (product number F8801 from Thermo Scientific) markers. These 100 nm large spheres have an excitation maximum of 580 nm. Fiducial markers were diluted 1000× into the water and then diluted 100× into PBS buffer (the final dilution of bead stock concentration was 100000×) and topped on the sample. Beads were let to settle for at least one hour. Just before imaging, the buffer was exchanged for the oxygen-scavenging buffer, and the sample was sealed. The final bead dilution was assessed by optimization to have a reasonable number of fiducial markers surrounding the imaged cells within the field of view. The text in the Methods section was updated accordingly.

The reviewer raises an important point regarding the inclusion/exclusion of the apparent cell edges in the analysis that would result in obvious artificial data bias. Indeed, the cell's edges were excluded from the analysis performed using the ClusterViSu. ClusterViSu offers a freehand selection drawing tool to define the region of interest included for the clustering analysis. This tool was used to define the region of interest that included the whole inner area of the cell – i.e., the whole cell, excluding the concentrated membrane regions prominent in the TIRF field, the cell edges. Occasionally, apparent areas of the cell that did not adhere well to the surface of the supporting glass (non-planar regions of the cell membrane, resulting in the “edging” effect in the TIRF field) were excluded from the analyzed region of interest as well. However, it is important to state that no “picking and choosing” of the region of interest area has been done; the majority of the inner surface of the cells was included in the analysis. The text in the Methods section was updated accordingly.